# The preprophase band-associated kinesin-14 OsKCH2 is a processive minus-end-directed microtubule motor

Kuo-Fu Tseng[1], Pan Wang[1,2], Yuh-Ru Julie Lee[3], Joel Bowen[4], Allison M. Gicking[1], Lijun Guo[2], Bo Liu [3] & Weihong Qiu [1,5]

In animals and fungi, cytoplasmic dynein is a processive minus-end-directed motor that plays dominant roles in various intracellular processes. In contrast, land plants lack cytoplasmic dynein but contain many minus-end-directed kinesin-14s. No plant kinesin-14 is known to produce processive motility as a homodimer. OsKCH2 is a plant-specific kinesin-14 with an N-terminal actin-binding domain and a central motor domain flanked by two predicted coiled-coils (CC1 and CC2). Here, we show that OsKCH2 specifically decorates preprophase band microtubules in vivo and transports actin filaments along microtubules in vitro. Importantly, OsKCH2 exhibits processive minus-end-directed motility on single microtubules as individual homodimers. We find that CC1, but not CC2, forms the coiled-coil to enable OsKCH2 dimerization. Instead, our results reveal that removing CC2 renders OsKCH2 a nonprocessive motor. Collectively, these results show that land plants have evolved unconventional kinesin-14 homodimers with inherent minus-end-directed processivity that may function to compensate for the loss of cytoplasmic dynein.

[1] Department of Physics, Oregon State University, Corvallis, OR 97331, USA. [2] Institute of Photobiophysics, Henan University, Kaifeng 475004 Henan, China. [3] Department of Plant Biology, University of California at Davis, Davis, CA 95616, USA. [4] Department of Mathematics, Oregon State University, Corvallis, OR 97331, USA. [5] Department of Biochemistry and Biophysics, Oregon State University, Corvallis, OR 97331, USA. These authors contributed equally: Kuo-Fu Tseng, Pan Wang, Yuh-Ru Julie Lee. Correspondence and requests for materials should be addressed to B.L. (email: BLiu@ucdavis.edu) or to W.Q. (email: Weihong.Qiu@physics.oregonstate.edu)

Kinesin and cytoplasmic dynein are both microtubule-based motor proteins that convert chemical energy from ATP hydrolysis into mechanical work for a variety of essential cellular processes[1–3]. Inherently processive microtubule-based motors are ones that are able to move continuously for several micrometers on single microtubules before dissociation without having to form multi-motor ensembles. Such motors are more likely to accomplish the same microtubule-based tasks with fewer motors[4]. Cytoplasmic dynein is the primary motor protein for cellular processes that depend on microtubule-based minus-end-directed motility in animal and fungal cells, which is usually attributed to its remarkable ability to generate processive minus-end-directed motility without clustering. In contrast, kinesin-14s are commonly nonprocessive minus-end-directed motor proteins[5–16]. Among all kinesin-14s studied to date, Kar3 from *Saccharomyces cerevisiae* and KlpA from *Aspergillus nidulans* are the only two known to have intrinsic processivity: Kar3 achieves processive motility toward the microtubule minus end by forming a heterodimer with the non-motor proteins Cik1 or Vik1[17, 18], and KlpA exhibits processive motility toward the microtubule plus end as a single homodimer[19].

Unlike animals and fungi, land plants have no cytoplasmic dynein in their motor protein repertoire but contain instead a large number of kinesin-14s[20]. For example, there are 21 kinesin-14s in *Arabidopsis thaliana*[21], 18 in *Oryza sativa*[22], and 15 in *Physcomitrella moss*[23]. Kinesins with a calponin homology domain (KCHs) are a distinct subset of kinesin-14s found exclusively in land plant cells[24]; there are at least 7 KCH proteins in *A. thaliana*[21], 9 in *O. sativa*[22], and 4 in *P. moss*[23]. KCH proteins likely function to mediate the crosstalk between the microtubule and actin networks inside plant cells, as the CH domain is often found in proteins that interact with the actin filaments (AFs)[25, 26]. Consistent with this notion, a number of KCHs have been shown to bind to microtubules and AFs simultaneously[27–32]. While it has long been speculated that land plants might have evolved unconventional kinesin-14s with intrinsic minus-end-directed processivity to compensate for the loss of cytoplasmic dynein[20, 33], no plant kinesin-14 has yet been found that exhibits processive minus-end-directed motility on single microtubules as individual homodimers[15, 34].

In this study, we report our systematic characterization of OsKCH2, a KCH protein from *O. sativa* that contains an N-terminal CH domain and a central microtubule-binding motor domain flanked by two predicted coiled-coils (CC1 and CC2). We find that OsKCH2 localizes to the microtubule bundles in the preprophase band at the prophase in vivo. Using total internal reflection fluorescence (TIRF) microscopy, we further show that purified OsKCH2 clusters to transport AFs along microtubules with minus-end-directed motility in vitro. Importantly, our single-molecule TIRF microscopy experiments reveal that unlike all other kinesin-14s that have been studied to date, OsKCH2 is a novel kinesin-14 motor that is capable of producing processive minus-end-directed motility on single microtubules as a homodimer. While only CC1 is necessary and sufficient to enable OsKCH2 dimerization, CC2 plays an indispensable role in OsKCH2 processivity and apparently enhances its binding to microtubules in vitro. Interestingly, CC2 substitution with that from the processive OsKCH2 is sufficient to enable OsKCH1—a nonprocessive homolog of OsKCH2—for processive minus-end-directed motility on single microtubules as a homodimer. Collectively, our results not only show that land plants have evolved unconventional kinesin-14 motors with intrinsic minus-end-directed processivity, but also markedly advance current knowledge of the design principles of kinesin-14s.

## Results

**OsKCH2 simultaneously interacts microtubules and AFs.** OsKCH2 is a KCH protein originally discovered in a genome-wide search for kinesins involved in the somatic cell division in cultured rice cells[22]. Two different rice KCH proteins were inadvertently given the same name OsKCH1 in two separate published works[22, 31]. We thus renamed the rice KCH from our previous study[22] to OsKCH2 in this study to avoid confusion. The full-length OsKCH2 consists of an actin-binding CH domain (aa 23–143), two upstream putative coiled-coils (CC0, aa 239–295; CC1, aa 313–354), a conserved neck motif (aa 370–383), a microtubule-binding motor domain (aa 384–711), a neck mimic (aa 712–717), a downstream putative coiled-coil (CC2, aa 718–766), and a C terminus (aa 768–1029) (Fig. 1a and Supplementary Fig. 1).

We performed the phylogenetic analysis of all kinesin-14s from *A. thaliana* and *O. sativa* (Supplementary Fig. 2 and Supplementary Data 1), which showed that OsKCH2 is more closely related to AtKP1, a KCH protein that was previously found to localize to mitochondria in *A. thaliana*[35]. We thus wanted to determine the intracellular localization of OsKCH2. To do that, we first generated an antibody against OsKCH2 for immunofluorescence, which revealed punctate signals at the cell cortex in a belt surrounding the prophase nucleus (Supplementary Fig. 3a). This signal was specific to OsKCH2, as no signal was detected in the TOS17 *kch2* mutant cells (Supplementary Fig. 3b). This localization pattern resembled the preprophase band (PPB)[36, 37], a plant-specific ring-shaped cortical structure that contains mainly microtubules and AFs and plays an essential role in division plane establishment[38, 39]. We next performed immuno-localization experiments to reveal the spatial relationship between OsKCH2 and the PPB microtubules. The results showed that the punctate OsKCH2 signal distributed along the PPB microtubules at prophase that could be viewed from different angles (Fig. 1b). No noticeable OsKCH2 signal was detected along microtubules of other arrays during mitosis, such as the spindles. We conclude that OsKCH2 localizes to the PPB microtubule array in a cell cycle-dependent manner.

It was shown recently that OsKCH1 molecules collectively transport AFs on the microtubules in the presence of ATP[16]. Given that the PPB also contains AFs, we next wanted to determine whether purified OsKCH2 was sufficient to transport AFs on the microtubule. To do that, we purified OsKCH2(1–767) —a truncated OsKCH2 that lacked the C terminus—for an in vitro actin transport assay (Fig. 1a, c, d). The assay showed that OsKCH2(1–767) transported AFs on single microtubules toward the minus ends (Fig. 1e and Supplementary Movie 1). Like OsKCH1[16], OsKCH2 also transported AFs with two distinct velocities ($V_{slow}$ and $V_{fast}$, Fig. 1f), which were determined to be $V_{slow} = 12 \pm 6$ nm s$^{-1}$ and $V_{fast} = 32 \pm 7$ nm s$^{-1}$ (mean ± s.d., $n = 231$, Fig. 1g). Collectively, these results suggest that OsKCH2 is an authentic KCH protein that is capable of interacting simultaneously with AFs and microtubules.

**OsKCH2(289-767) is a processive minus-end-directed motor.** We next characterized the motility of GFP-OsKCH2(289–767), a truncated motor-neck construct that contains two putative coiled-coils CC1 and CC2 sandwiching the motor domain (Fig. 2a, b); it is worth emphasizing that GFP-OsKCH2 (289–767) lacks the N-terminal CH domain, the other putative coiled-coil CC0 before CC1, and the C terminus. We first performed an ensemble microtubule gliding assay to determine its directionality (Fig. 2c). Briefly, GFP-OsKCH2(289–767) molecules were anchored on the coverslip via an N-terminal polyhistidine-tag, and the directionality of GFP-OsKCH2

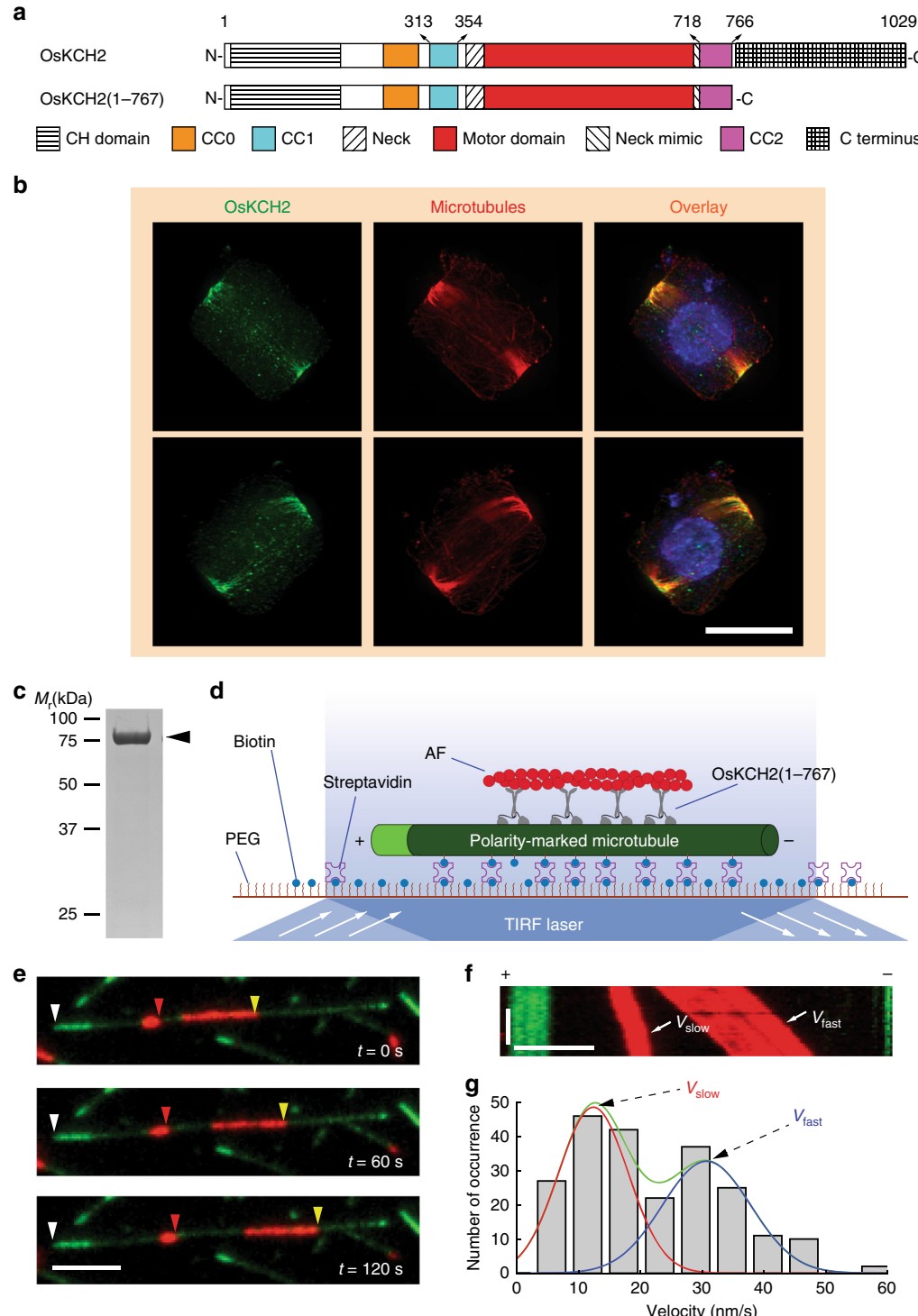

**Fig. 1** OsKCH2 localizes to the PPB at prophase in vivo and transports AFs on the microtubule with minus-end-directed motility in vitro. **a** Schematic diagrams of the full-length OsKCH2 and OsKCH2(1–767). **b** OsKCH2 shows a punctate localization pattern along the PPB microtubules at prophase. Top and bottom rows are triple labeling of OsKCH2 (green), microtubules (red), and the nucleus (blue) in a rice cell at prophase when viewed from two different angles. **c** Coomassie-stained SDS-polyacrylamide gel electrophoresis (SDS-PAGE) of purified recombinant OsKCH2(1–767). **d** Schematic diagram of the AF transport assay. **e** Micrograph montage showing that OsKCH2(1–767) transports rhodamine-labeled AFs (red) along an Alexa 488-labeled polarity-marked microtubule (green) toward the minus end. White arrowheads indicate the microtubule plus end, and red and yellow arrowheads indicate the leading ends of two different AFs. **f** Kymograph of two AFs shown in **e** moving at a fast velocity ($V_{fast}$) and a slow one ($V_{slow}$). **g** Velocity histogram of AF transport along microtubules with two distinct velocities. The velocity histogram was fitted to a combination of two Gaussian distributions. The green curve indicates the overall fit, and red and blue curves indicate the slow and fast velocity distribution, respectively. Scale bars: 1 min (vertical) and 5 μm (horizontal)

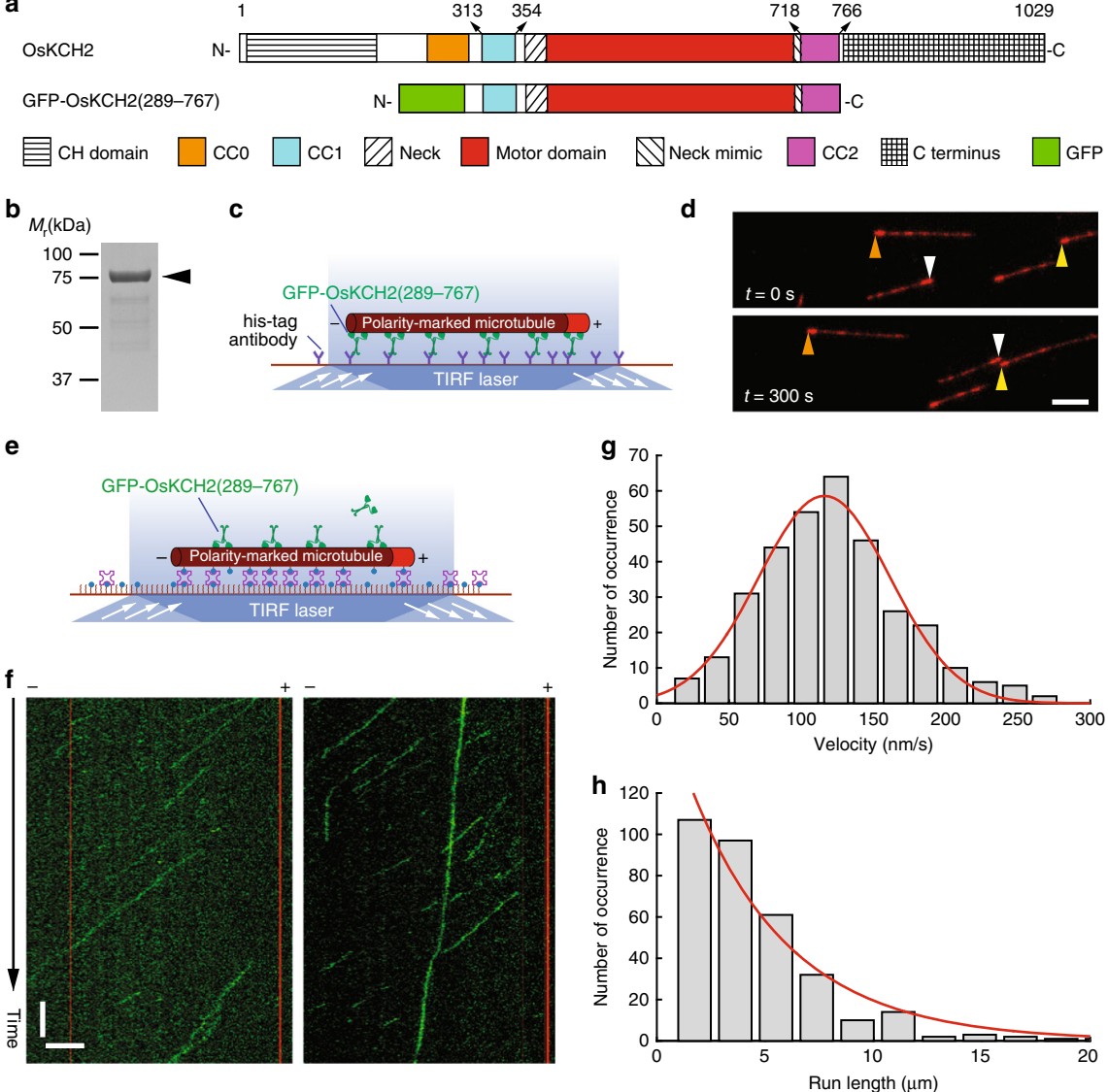

**Fig. 2** GFP-OsKCH2(289-767) moves on the microtubule with minus-end-directed processive motility. **a** Schematic diagrams of the full-length OsKCH2 and GFP-OsKCH2(289–767). **b** Coomassie-stained SDS-PAGE of purified recombinant GFP-OsKCH2(289–767). **c** Schematic diagram of the ensemble microtubule gliding assay. **d** Micrograph montage showing the motion of polarity-marked microtubules (red) glided by surface-immobilized GFP-OsKCH2 (289–767). Arrowheads (brown, white, and yellow) indicate the plus ends of three different microtubules. **e** Schematic diagram of the single-molecule motility assay for observing the movement of individual GFP-OsKCH2(289–767) molecules on single microtubules. **f** Kymographs of single GFP-OsKCH2 (289–767) molecules (green) moving processively toward the minus end of single polarity-marked microtubules (red). **g** Velocity histogram of single GFP-OsKCH2(289–767) molecules. Red line indicates a Gaussian fit to the velocity histogram. **h** Run-length histogram of single GFP-OsKCH2(289–767) molecules. Red line indicates a single-exponential fit to the run-length histogram. Scale bars: 1 min (vertical) and 5 μm (horizontal)

(289–767) was deduced from the motion of polarity-marked microtubules. The assay showed that GFP-OsKCH2(289–767) behaved like a minus-end-directed kinesin motor, as polarity-marked microtubules were driven to move on the coverslip surface with the brightly labeled plus ends leading (Fig. 2d and Supplementary Movie 2). To directly observe the motility of individual GFP-OsKCH2(289–767) molecules on surface-immobilized polarity-marked microtubules, we performed a single-molecule motility assay (Fig. 2e). The assay showed that, contrary to the notion that kinesin-14s are commonly non-processive minus-end-directed motors, GFP-OsKCH2 (289–767) unexpectedly moved on single microtubules in a processive manner toward the minus ends (Fig. 2f and Supplementary Movies 3 and 4). Quantitative kymograph analysis of the motility of GFP-OsKCH2(289–767) revealed a velocity

of $115 \pm 58$ nm s$^{-1}$ (mean ± s.d., $n = 332$, Fig. 2g) and a characteristic run length of $4.6 \pm 0.5$ μm (mean ± s.e.m., $n = 332$, Fig. 2h).

Recent studies have shown that as few as two nonprocessive kinesin-14 molecules could couple to achieve minus-end-directed processive motility on single microtubules[15, 40]. To rule out that GFP-OsKCH2(289–767) achieved processive minus-end-directed motility by inadvertently forming high-order oligomers, we conducted several additional experiments. First, we performed single-molecule photobleaching and sucrose gradient centrifugation assays to determine the oligomerization of GFP-OsKCH2 (289–767). The photobleaching assay showed that GFP-OsKCH2 (289–767) existed primarily as a homodimer in solution, as it was photobleached in a pattern similar to other dimeric kinesins[15] and predominantly with a single step or two steps (Fig. 3a, b),

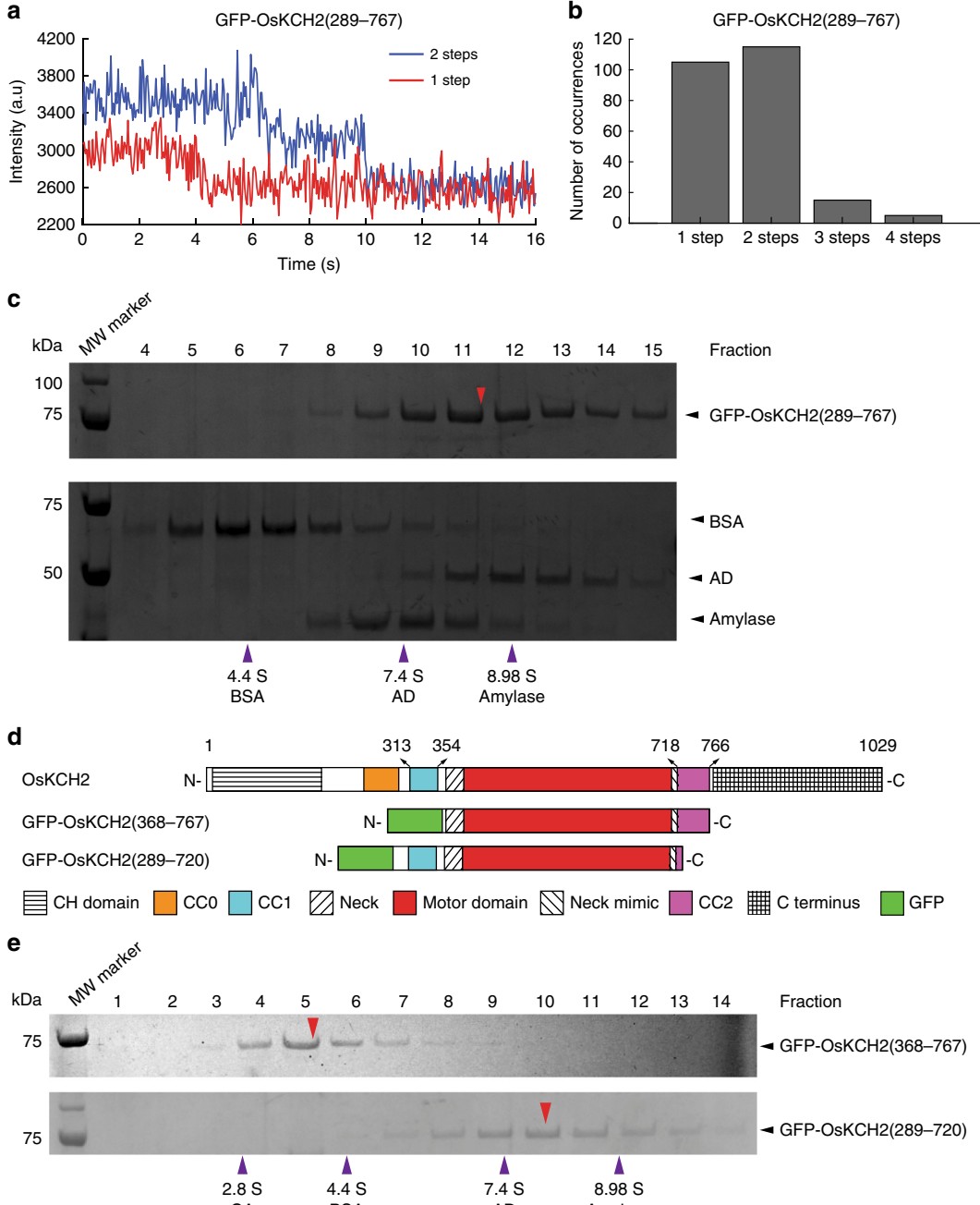

**Fig. 3** GFP-OsKCH2(289-767) forms a homodimer via CC1. **a** Example fluorescence intensity traces over time of individual GFP-OsKCH2(289–767) molecules immobilized on the microtubules. **b** Histogram of the photobleaching steps of GFP-OsKCH2(289–767) ($n = 240$). **c** SDS-PAGE analysis of fractions from the sucrose gradient centrifugation assays of GFP-OsKCH2(289–767) (top) and the standard proteins (bottom). For each protein, the band intensity distribution was fit to a Gaussian function to determine the peak fraction. Red arrowhead corresponds to the estimated peak fraction position of GFP-OsKCH2(289–767); and purple arrowheads correspond to the estimated peak fraction positions of the standard proteins as indicated. The molecular weight of GFP-OsKCH2(289–767) was estimated to be $177.9 \pm 3.8$ kDa (mean $\pm$ s.d., $n = 3$). **d** Schematic diagrams of the full-length OsKCH2, GFP-OsKCH2(368–767), and GFP-OsKCH2(289–720). **e** SDS-PAGE analysis of fractions from the sucrose gradient centrifugation assays of GFP-OsKCH2 (368–767) (top) and GFP-OsKCH2(289–720) (bottom). Red arrowheads correspond to the peak fraction positions of GFP-OsKCH2(368–767) and GFP-OsKCH2(289–720); and purple arrowheads correspond to the peak fraction positions of the standard proteins as indicated. For each protein, the peak fraction was estimated by fitting the band intensity distribution to a Gaussian function. The molecular weights of GFP-OsKCH2(368–767) and GFP-OsKCH2(289–720) were determined to be $60.9 \pm 7.9$ kDa and $155.7 \pm 8.3$ kDa, respectively (mean $\pm$ s.d., $n = 3$)

Consistent with the photobleaching assay, the sucrose gradient assay showed that GFP-OsKCH2(289–767) migrated with a mean sedimentation coefficient of 8.26 S (Fig. 3c), which yielded a molecular weight (MW = ~178 kDa) approximately twice that of a monomeric GFP-OsKCH2(289–767) (MW = 82.1 kDa).

Second, we modified GFP-OsKCH2(289–767) to generate GFP-OsKCH2(289–767)$^{T}$, which contains an insertion of the coding sequence for a GCN4 parallel tetramer motif[41] between GFP and OsKCH2 (Supplementary Fig. 4a). In a recent study[15], Johnson and colleagues made an artificial kinesin14-VIb homotetramer

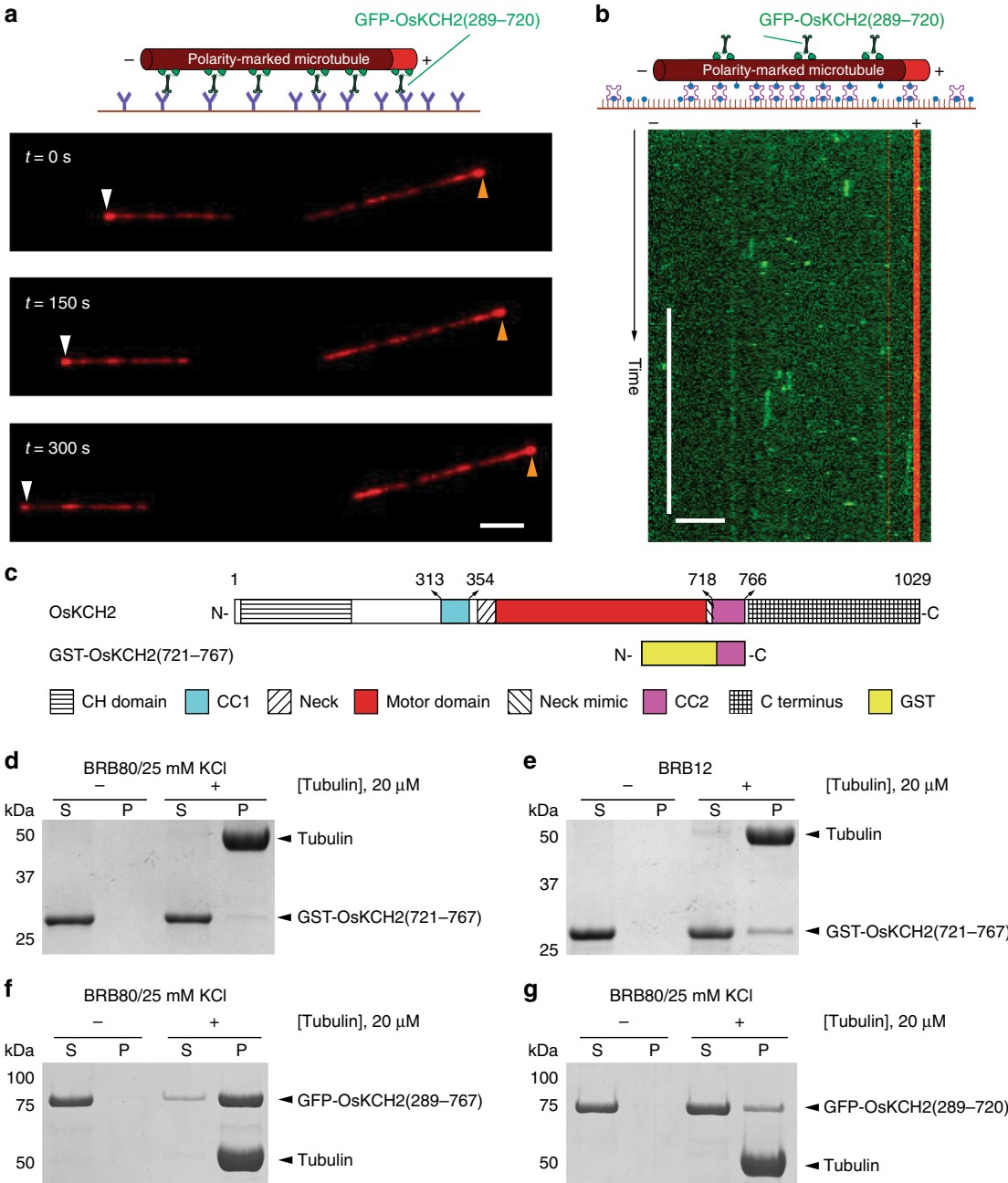

**Fig. 4** CC2 enables GFP-OsKCH2(289-767) processivity by enhancing its microtubule-binding affinity. **a** Micrograph montage showing the gliding motion of polarity-marked microtubules (red) driven by surface-immobilized GFP-OsKCH2(289–720). White and brown arrowheads indicate the plus end of a polarity-marked microtubule at three different time points. **b** Example kymograph showing the nonprocessive motility of GFP-OsKCH2(289–720) on the microtubule. **c** Schematic diagrams of the full-length OsKCH2 and GST-OsKCH2(721–767). **d** Coomassie-stained SDS-PAGE of the microtubule co-sedimentation assay for GST-OsKCH2(721–767) in BRB80/25 mM KCl. **e** Coomassie-stained SDS-PAGE of the microtubule co-sedimentation assay for GST-OsKCH2(721–767) in BRB12. **f** Coomassie-stained SDS-PAGE of the microtubule co-sedimentation assay for GFP-OsKCH2(289–767) in BRB80/25 mM KCl. **g** Coomassie-stained SDS-PAGE of the microtubule co-sedimentation assay for GFP-OsKCH2(289–720) in BRB80/25 mM KCl. Scale bars: 30 s (vertical) and 5 μm (horizontal)

composed of two identical dimers using the same GCN4 tetramer motif. Similar to the engineered kinesin14-VIb homotetramer[15], the photobleaching pattern of GFP-OsKCH2(289–767)$^T$ differed drastically from that of GFP-OsKCH2(289–767) and contained a high percentage of 3- and 4-step photobleaching processes with the peak at the 3-step photobleaching (Supplementary Fig. 4b, c), indicating that OsKCH2(289–767)$^T$ formed a homotetramer in solution. Our single-molecule motility experiments showed that OsKCH2(289–767)$^T$ moved processively toward the minus ends

on single microtubules (Supplementary Fig. 4d and Supplementary Movie 5). Quantitative kymograph analysis of the motility of GFP-OsKCH2(289–767)$^T$ revealed a velocity of $100 \pm 49$ nm s$^{-1}$ (mean ± s.d., $n = 287$, Supplementary Fig. 4e) and a characteristic run length of $16.5 \pm 3.0$ μm (mean ± s.e.m., $n = 287$, Supplementary Fig. 4f). While the velocity of GFP-OsKCH2(289–767)$^T$ was nearly the same as that of GFP-OsKCH2(289–767), its run length was >3.5 times longer than that of GFP-OsKCH2(289–767). Collectively, these results show that the observed minus-end-

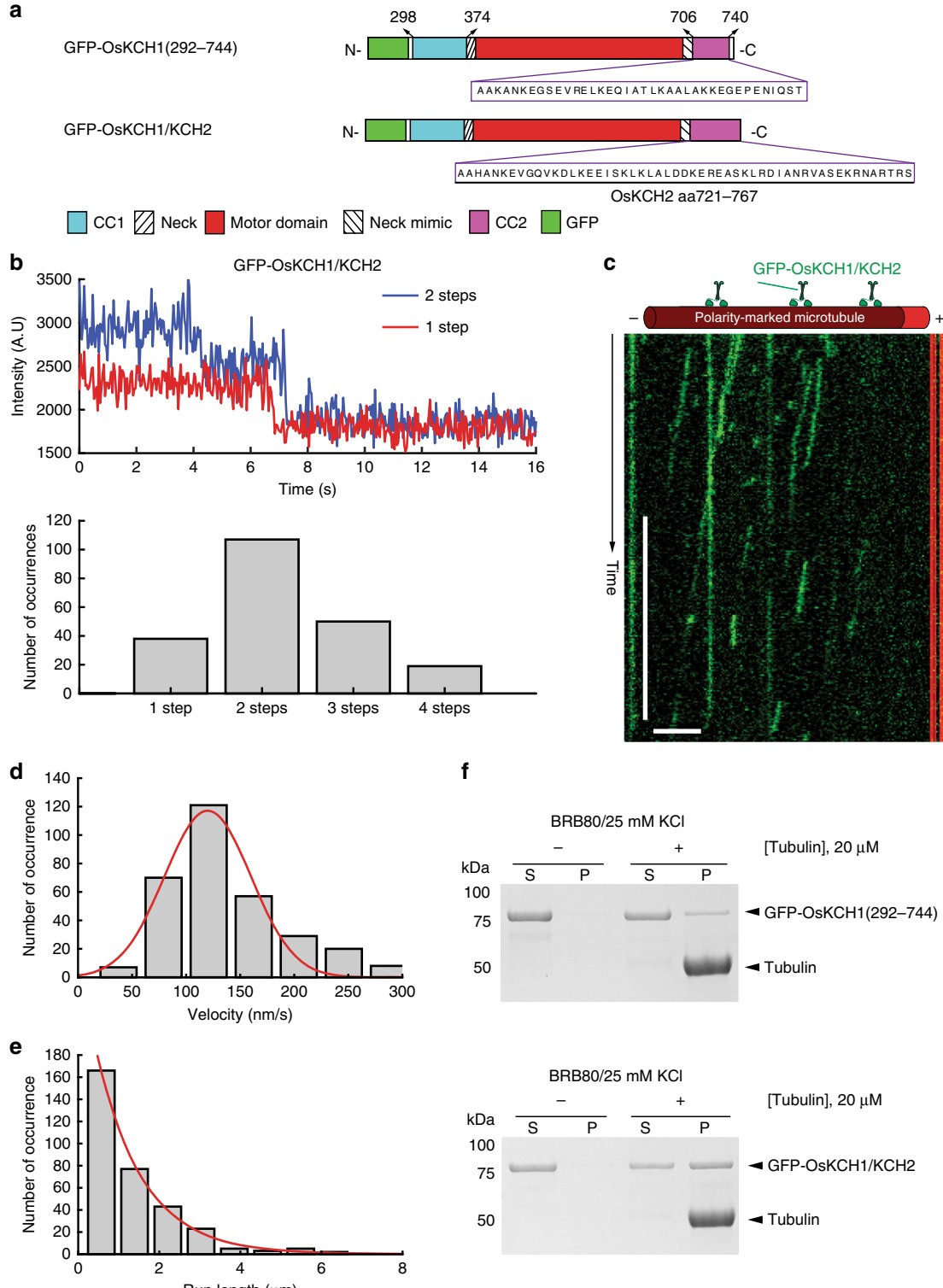

**Fig. 5** The GFP-OsKCH1/KCH2 chimera is a processive minus-end-directed motor. **a** Schematic diagrams of GFP-OsKCH1(292–744) and the GFP-OsKCH1/KCH2 chimera. GFP-OsKCH1/KCH2 is a derivative of GFP-OsKCH1(292–744) by replacing its endogenous CC2 with that from OsKCH2. **b** Photobleaching analyses of GFP-OsKCH1/KCH2. (Top) Example fluorescence intensity traces over time of individual GFP-OsKCH1/KCH2 molecules immobilized on the microtubules. (Bottom) Histogram of the photobleaching steps of GFP-OsKCH1/KCH2 ($n = 214$). **c** Example kymograph of individual GFP-OsKCH1/KCH2 molecules (green) moving processively toward the minus end of single polarity-marked microtubules (red). **d** Velocity histogram of single GFP-OsKCH1/KCH2 molecules. Red line indicates a Gaussian fit to the velocity histogram. **e** Run-length histogram of single GFP-OsKCH1/KCH2 molecules. Red line indicates a single-exponential fit to the run-length histogram. **f** Coomassie-stained SDS-PAGE of the microtubule co-sedimentation assay for (top) GFP-OsKCH1(292–744) and (bottom) GFP-OsKCH1/KCH2 in BRB80/25 mM KCl. Scale bars: 30 s (vertical) and 5 μm (horizontal)

directed processive motility for GFP-OsKCH2(289–767) is not due to inadvertent motor coupling and/or clustering, but rather is an intrinsic behavior of the motor as a homodimer.

**OsKCH2 is dimerized via the upstream coiled-coil CC1**. The motor domain of GFP-OsKCH2(289–767) is flanked by a pair of putative coiled-coils, CC1 and CC2 (Fig. 2a). We thus wanted to ascertain whether CC1 and CC2 both are required for forming the GFP-OsKCH2(289–767) homodimer. To address this, we engineered GFP-OsKCH2(368–767) and GFP-OsKCH2(289–720) lacking CC1 and CC2, respectively (Fig. 3d), and performed the sucrose gradient centrifugation assay to determine their oligomerization. The assay showed that the CC1-less GFP-OsKCH2(368–767) migrated as a monomer with a mean sedimentation coefficient of 4.14 S, whereas the CC2-less GFP-OsKCH2(289–720) migrated as a dimer with a mean sedimentation coefficient of 7.52 S (Fig. 3e). Thus, CC1 but not CC2 is both necessary and sufficient for the formation of an GFP-OsKCH2(289-767) homodimer.

**The processive motility of OsKCH2(289-767) is enabled by CC2**. As an additional control of our motility assays, we also determined the motility of GFP-OsKCH1(292–744) (Supplementary Fig. 5a–c). GFP-OsKCH1(292–744)—an OsKCH1 equivalent of GFP-OsKCH2(289–767)—was recently shown to be a nonprocessive minus-end-directed kinesin-14 motor[16]. In agreement with that result, our measurements showed that GFP-OsKCH1(292–744) formed a homodimer (Supplementary Fig. 5d, e), exhibited minus-end-directed motility in the ensemble microtubule gliding assay (Supplementary Fig. 5f and Supplementary Movie 6), and behaved like a nonprocessive kinesin in the single-molecule motility assay (Supplementary Fig. 5g and Supplementary Movie 7).

The drastic difference in processivity between GFP-OsKCH1(292–744) and GFP-OsKCH2(289–767) prompted us to perform a pairwise protein sequence alignment between OsKCH1(292–744) and OsKCH2(289–767), which showed that these two constructs differed most markedly from each other in the CC2 region (aa 718–766 in OsKCH2 and aa 706–740 in OsKCH1) with CC2 from OsKCH2(289–767) containing a few more positively charged residues (Supplementary Fig. 6). To determine whether CC2 plays a role in the processive motility of GFP-OsKCH2(289–767), we characterized the motility of the CC2-less GFP-OsKCH2(289–720) construct (Fig. 3d). Like GFP-OsKCH2(289–767), GFP-OsKCH2(289–720) exhibited minus-end-directed motility in the microtubule gliding assay (Fig. 4a and Supplementary Movie 8). However, in contrast to GFP-OsKCH2(289–767) and similar to other nonprocessive kinesin-14s[15, 16, 42], GFP-OsKCH2(289–720) was unable to exhibit processive motility and only transiently interacted with the microtubule (Fig. 4b and Supplementary Movie 9). These results show that CC2 is required for OsKCH2(289–767) to achieve processive motility on single microtubules.

Several kinesins are known to achieve processive motility on single microtubules via non-motor microtubule-binding domains[17, 19, 43]. Given that CC2 in OsKCH2 contains quite a number of positively charged residues (Supplementary Fig. 6), we wanted to determine whether CC2 contains the ability to independently bind to microtubules. To do that, we created a fusion protein GST-OsKCH2(721–767) containing the majority of the CC2 region for an in vitro microtubule co-sedimentation assay (Fig. 4c). The assay showed that purified CC2 only weakly interacted with the microtubule in both high and low ionic strength buffer conditions (Fig. 4d, e). However, when we performed a microtubule co-sedimentation assay to compare the

microtubule binding of GFP-OsKCH2(289–767) and GFP-OsKCH2(289–720), we found that GFP-OsKCH2(289–767) bound to the microtubule much more strongly than GFP-OsKCH2(289–720) (Fig. 4f, g). Based on these observations, we additionally engineered a quadruple mutant of the processive GFP-OsKCH2(289–767) to replace the last four positively charged residues to the neutral alanine (GFP-K760A/R761A/R764A/R766A, Supplementary Fig. 7a). We found that the quadruple mutant retained the ability to collectively glide microtubules with minus-end-directed motility (Supplementary Fig. 7b and Supplementary Movie 10), but lacked the ability to move processively on single microtubules (Supplementary Fig. 7c and Supplementary Movie 11) and displayed significantly reduced microtubule-binding affinity (Supplementary Fig. 7d) compared to the processive GFP-OsKCH2(289–767) (Fig. 4f). Collectively, these results show that CC2 enables OsKCH2(289–767) for processive motility on the microtubule likely by enhancing its microtubule-binding affinity via those positively charged residues.

**CC2 substitution enables OsKCH1 for processive motility**. We next asked how the motility of OsKCH1 would be affected when its CC2 is substituted with that from OsKCH2. To that end, we engineered GFP-OsKCH1/KCH2, a chimera derived from GFP-OsKCH1(292–744) by replacing its CC2 with that of OsKCH2 (Fig. 5a). Like GFP-OsKCH1(292–744), GFP-OsKCH1/KCH2 formed a homodimer (Fig. 5b), and exhibited minus-end-directed motility in the ensemble microtubule gliding assay (Supplementary Fig. 8 and Supplementary Movie 12). However, unlike GFP-OsKCH1(292–744), which lacks the ability to move processively on single microtubules as a homodimer (Supplementary Fig. 5g and Supplementary Movie 7), GPF-OsKCH1/KCH2 surprisingly was observed to move in a processive manner toward the minus ends on single microtubules (Fig. 5c and Supplementary Movie 13). Quantitative kymograph analysis of the GFP-OsKCH1/KCH2 motility revealed a velocity of $120 \pm 40$ nm s$^{-1}$ (mean ± s.d., $n = 324$, Fig. 5d) and a characteristic run length of $1.15 \pm 0.05$ µm (mean ± s.e.m., $n = 324$, Fig. 5e). We also performed a microtubule co-sedimentation assay to compare the microtubule binding of GFP-OsKCH1(292–744) and GFP-OsKCH1/KCH2, and the results showed that GFP-OsKCH1(292–744) bound to the microtubule much more weakly than the GFP-OsKCH1/KCH2 chimera (Fig. 5f).

**Discussion**

To summarize, we have revealed an unexpected finding that OsKCH2(289–767), a motor-neck construct of the kinesin-14 OsKCH2 from the rice plant *O. sativa*, is a novel processive minus-end-directed microtubule motor. To our knowledge, OsKCH2(289–767) is the first homodimeric kinesin-14 to demonstrate processive minus-end-directed motility on single microtubules without clustering. This finding markedly expands our knowledge of the diversified design principles of kinesin-14s. Importantly, this study shows that some land plants, if not all, have evolved unconventional kinesin-14s with intrinsic minus-end-directed processive motility, which could potentially function to compensate for the lack of cytoplasmic dynein[20, 33].

We have further shown that the putative coiled-coil region CC2 (aa 718–766) does not form an authentic coiled-coil to contribute to the dimerization of GFP-OsKCH2(289–767). Instead, our results show that CC2 apparently plays an indispensable role in its processive motility on single microtubules, as GFP-OsKCH2(289–720)—a truncation construct that lacks CC2—still forms a dimer (Fig. 3e) but fails to exhibit processive motility on the microtubule (Fig. 4b). How does CC2 function to

contribute to the processivity motility of OsKCH2? One possibility is that CC2 functions as an ATP-independent microtubule-binding site. In this case, while the motor domain and CC2 of OsKCH2 individually exhibit weak binding to microtubules (Fig. 4d, g), they are able to synergistically achieve tighter microtubule binding in GFP-OsKCH2(289–767) to promote processivity. Alternatively, CC2 does not act as an ATP-independent microtubule-binding domain, but instead interacts with the motor domain of GFP-OsKCH2(289–767) to enhance its interaction with the microtubule. This would be similar to how the microtubule binding of the *Arabidopsis* kinesin-14 AtKCBP is affected by the neck mimic outside its motor domain[44, 45]. Given that mutating the last four positively charged residues (K760, R761, R764, and R766) in CC2 altogether to alanine renders GFP-OsKCH2(289–767) a nonprocessive motor on single microtubules and that replacing CC2 in the nonprocessive GFP-OsKCH1(292–744) with that from OsKCH2 results in a processive GFP-OsKCH1/KCH2 (Fig. 5), we are in favor of the former notion that CC2 enables GFP-OsKCH2(289–767) for processive motility by acting as an ATP-independent microtubule-binding site. Future high-resolution cryo-EM structures of OsKCH2 constructs on single microtubules are needed to clarify the precise underlying biophysical mechanism of CC2-enabled processivity.

What are the implications of the processive motility of GFP-OsKCH2(289–767)? The PPB plays a critical role in cell division plane determination in flowering plants, although the exact mechanism is still unclear[39]. AFs are associated with the PPB microtubule bundles at prophase when plant cells undergo mitosis[46–48], but disappear prior to the disassembly of the PPB[49]. The transient presence of AFs at the PPB suggests that AF dynamics is regulated during the assembly and disassembly of the PPB. In this study, we show that OsKCH2 decorates the PPB microtubules at prophase in vivo (Fig. 1b) and clusters to transport AFs on the microtubules with minus-end-directed motility in vitro (Fig. 1e and Supplementary Movie 1). These results suggest that one likely function of OsKCH2 is to recruit AFs to the PPB by either dynamically translocating AFs on the PPB microtubules or statically crosslinking these two cytoskeletal filaments inside the PPB. In addition, OsKCH2 may function in vivo to control nuclear positioning in a way analogous to how cytoplasmic dynein moves the spindle and nucleus in budding yeast[50]: for example, to position the nucleus inside the cell, OsKCH2 becomes cortically anchored to the actin network via a mechanism involving the CH domain, captures the plus ends of cytoplasmic microtubules (that are attached to the nucleus via the minus ends), and uses its minus-end-directed motility to pull these cytoplasmic microtubules along the cortex. The ability of OsKCH2 to generate processive motility without clustering implies that OsKCH2 can keep a tight grip on the microtubule filament when carrying out its cellular function(s) and thus is able to achieve the same cellular task(s) with fewer motors. Systematic in vivo studies of the wild-type OsKCH2 and its processivity-deficient mutants will be important next steps toward revealing its cellular functions, the physiological role of the CC2 region, and the biological relevance of its processivity. Given that among all kinesin-14 motors from *A. thaliana* and *O. sativa*, OsKCH2 is more related to the *A. thaliana* KCH protein AtKP1, it would be interesting to investigate whether AtKP1 retained inherent minus-end-directed processive microtubule-based motility during the speciation process.

## Methods

**Molecular cloning of recombinant KCH constructs**. The full-length cDNAs of OsKCH1 and OsKCH2 were both codon optimized and synthesized for protein expression and purification from *E. coli*. All KCH constructs were integrated in a modified pET17b vector via isothermal assembly and verified by DNA sequencing.

Except for the GST-tagged CC2 construct GST-OsKCH2(721–767), all other constructs contained an N-terminal His-tag.

**Protein expression and purification**. For protein expression, plasmids were transformed into the BL21 Rosetta (DE3) competent cells (Novagen). Cells were grown at 37 °C in TPM (containing 20 g tryptone, 15 g yeast extract, 8 g NaCl, 2 g $Na_2HPO_4$, and 1 g $KH_2PO_4$ per 1 l) supplemented with 50 µg ml$^{-1}$ ampicillin and 30 µg ml$^{-1}$ chloramphenicol. Expression was induced by cold shock on ice at OD = 0.8–1 with 0.1 mM IPTG, and incubation was continued for additional 12–14 h at 20 °C. Cell pellets were harvested by centrifugation at 5500 × g for 10 min using a S-5.1 rotor (Beckman Coulter), and stored at −80 °C prior to cell lysis.

To purify His-tagged OsKCH1 and OsKCH2 motor constructs—GFP-OsKCH1 (292–744), OsKCH2(1–767), GFP-OsKCH2(289–767), GFP-OsKCH2(289–767)$^T$, GFP-OsKCH2(289–720), GFP-OsKCH2(368–767), GFP-K760A/R761A/R764A/R766A, and GFP-OsKCH1/KCH2—cell pellets were re-suspended in the lysis buffer (50 mM sodium phosphate buffer, pH 8.0, 250 mM NaCl, 1 mM $MgCl_2$, 0.5 mM ATP, 10 mM ß-mercaptoethanol, 20 mM imidazole, and 1 µg ml$^{-1}$ leupeptin, 1 µg ml$^{-1}$ pepstatin, 1 mM PMSF, and 5% glycerol), and lysed via sonication (Branson Sonifier 450). The cell lysate was then centrifuged at 21,000 × g for 30 min using a 75 Ti rotor (Beckman Coulter). The supernatant was incubated with Talon beads (Clontech) by end-to-end mixing at 4 °C for 1 h. The protein/beads slurry was then applied to a Poly-Prep column (Bio-Rad) and washed twice with 10 column volumes of the wash buffer (50 mM sodium phosphate buffer, pH 8.0, 250 mM NaCl, 1 mM $MgCl_2$, 0.1 mM ATP, 10 mM ß-mercaptoethanol, 20 mM imidazole, and 1 µg ml$^{-1}$ leupeptin, 1 µg ml$^{-1}$ pepstatin, 1 mM PMSF, and 5% glycerol). The protein was eluted with 5 column volumes of the elution buffer (50 mM sodium phosphate buffer, pH 8.0, 250 mM NaCl, 1 mM $MgCl_2$, 0.5 mM ATP, 10 mM ß-mercaptoethanol, 250 mM imidazole, and 5% glycerol). The eluted protein was flash frozen in liquid nitrogen, and stored at −80 °C. The GST-tagged CC2 construct GST-OsKCH2(721–767) was expressed and purified similarly to the His-tagged motor constructs except that the supernatant was incubated with 1 ml glutathione beads (Clontech) and the proteins were eluted with an elution buffer containing 10 mM glutathione.

**Production of anti-OsKCH2 antibodies and immunofluorescence**. The cDNA fragment encoding the OsKCH2 polypeptide of amino acids 171–313 was amplified using the primers of 5′-AAG ACC ATG GCT TCA TAT TCA TCC AGG G-3′ and 5′-AAG TGA GCT CCC ATC ATC CAA TTG TTT G-3′. The resulting fragment was digested with the enzymes NcoI and SacI and cloned into the pGEX-KG vector[51] at the identical sites. A GST fusion protein was then expressed in bacteria after the recombinant plasmid was introduced into the bacterial strain BL21(DE3) and purified by affinity chromatography using reduced glutathione resin (ThermoFisher Scientific, Catalog #25236). The purified fusion protein was used as an antigen for immunization in rabbits at the Comparative Pathology Laboratory, University of California in Davis, where the antibody production protocol was approved by the UC Davis Institutional Animal Care and Use Committee (IACUC). OsKCH2 antibodies were purified using columns of GST and GST-OsKCH2(171–313) proteins, which had been, respectively, immobilized on AminoLink$^{TM}$ Plus coupling resins (ThermoFisher Scientific, Catalog #20505) according to the manufacturer's instruction. Anti-GST antibodies were first depleted from the antisera by the GST column. The flow through was then applied to the GST-OsKCH2(171–313) column. Specific antibodies against OsKCH2 were eluted from the GST-OsKCH2(171–313) column with 100 mM glycine (pH 2.5) and immediately neutralized with 1/10 volume of 1 M Tris-HCl (pH 8.0)[52].

Root tips were excised from 7-day-old rice seedlings grown in dark. They were fixed for 1 h in freshly prepared 4% formaldehyde in PME (0.05 M PIPES buffer, pH 6.9, 5 mM $MgSO_4$, and 1 mM EGTA) at room temperature, followed by a treatment of 1% (w/v) Cellulase RS (Yakult Pharmaceutical, Catalog #203042) for 30 min. Root tip cells were treated with 0.5% (v/v) Triton X-100 for 15 min prior to antibody incubation[53]. In addition to the detection of OsKCH2 with purified anti-OsKCH2 antibodies (1:200 diluted in PBS, 140 mM NaCl, 2.7 mM KCl, 10 mM $Na_2HPO_4$, and 1.8 mM $KH_2PO_4$, pH 7.3), microtubules were labeled with the DM1A monoclonal anti-α-tubulin antibody (Sigma-Aldrich 9026, 1:400) for 3 h at room temperature. Following rinses with PBS, the cells were incubated for 1 h at room temperature with secondary antibodies, Texas Red-conjugated donkey anti-mouse IgG (Rockland Immunochemicals, Catalog #610-709-124; 1:200) and fluorescein isothiocyanate (FITC)-conjugated donkey anti-rabbit IgG antibodies (Rockland Immunochemicals, Catalog #611-702-127; 1:200). Nuclei were stained with 4′,6′-diamine-2-phenylindole (DAPI) included in the mounting medium ProLong$^{TM}$ (ThermoFisher Scientific, Catalog #P36966). Results were recorded using a DeltaVision microscope (Applied Precision) using standard settings for FITC and Texas Red.

**Taxol-stabilized microtubules**. Taxol-stabilized polarity-marked microtubules with bright plus ends were prepared as previously described[54]. To make the polarity-marked microtubules, a dim tubulin mix [containing 17 µM unlabeled tubulin and 0.8 µM tetramethylrhodamine- (TMR) or HiLyte647-tubulin] was first incubated in BRB80 with 0.5 mM GMPCPP (Jena Bioscience) at 37 °C for 2 h to make dim microtubules, and then centrifuged at 250,000 × g for 7 min at 37 °C in a

TLA100 rotor (Beckman Coulter). The pellet was re-suspended in a bright tubulin mix (containing 7.5 μM unlabeled tubulin, 4 μM TMR- or HiLyte647-tubulin, and 15 μM NEM-tubulin) in BRB80 with 2 mM GMPCPP and incubated at 37 °C for additional 15 min to cap the plus end of the dim microtubules. The resulting polarity-marked track microtubules were pelleted at $20,000 \times g$ for 7 min at 37 °C in the TLA100 rotor (Beckman Coulter), and finally re-suspended in BRB80 with 40 μM taxol. For making track microtubules, the dim tubulin mix also contained additional 17 μM biotinylated tubulin.

**Preparation of fluorescent actin filaments.** The actin stock solution (10 mg ml$^{-1}$) was reconstituted in the general actin buffer (GAB, 5 mM Tris-HCl, pH 8, 0.2 mM CaCl$_2$, 1 mM DTT, and 0.2 mM ATP) from the lyophilized actin (Cytoskeleton). For making actin filaments, 2 μl of the actin stock was further diluted in the general actin buffer to a final concentration of 1 mg ml$^{-1}$ and kept on ice for 1 h to depolymerize all filaments. After 1-h incubation on ice, 2 μl of 10× polymerization buffer (100 mM Tris-HCl, pH 7.5, 500 mM KCl, 20 mM MgCl$_2$, 10 mM ATP) was added to the 20 μl actin solution and kept at room temperature to polymerize actin filaments. The polymerized actin filament was then diluted in 1× polymerization buffer supplemented with 70 nM rhodamine–phalloidin (Invitrogen) to label the actin filaments.

**TIRF microscopy experiments.** All time-lapse imaging assays were performed at room temperature using the Axio Observer Z1 objective-type TIRF microscope (Zeiss) equipped with a ×100 1.46 NA oil-immersion objective and a back-thinned electron multiplier CCD camera (Photometrics). All microscope coverslips were functionalized with biotin-PEG as previously described[55] to reduce nonspecific surface absorption of molecules[55]. All time-lapse imaging experiments in this study used flow chambers that were made by attaching a functionalized coverslip to a microscope glass slide by double-sided tape as previously described[56].

For the actin transport assay, the flow chamber was incubated with one chamber volume of 0.5 mg ml$^{-1}$ of streptavidin for 2 min at room temperature, washed with two chamber volumes of BRB15 (15 mM PIPES, pH 6.8, 1 mM MgCl$_2$, and 1 mM EGTA) supplemented with 20 μM taxol to remove excessive unbound streptavidin, and incubated with one chamber volume of taxol-stabilized polarity-marked Alexa 488/Biotin-labeled microtubules at room temperature for 5 min. Unbound microtubules were removed by washing the chamber with five chamber volumes of BRB80 supplemented with 20 μM taxol and 1.3 mg ml$^{-1}$ casein. The flow chamber was next perfused with OsKCH2(1–767) diluted in BRB80 (80 mM PIPES, pH 6.8, 1 mM EGTA, and 1 mM MgCl$_2$) supplemented with 20 μM taxol, and 1.3 mg ml$^{-1}$ casein, incubated at room temperature for 2 min, and washed with five chamber volumes of the same BRB80-based buffer to remove unbound OsKCH2(1–767) motors. The chamber was then perfused with rhodamine-labeled actin filaments, incubated at room temperature for 2 min, and washed with the same BRB80-based buffer to remove unbound actin filaments. Finally, the flow chamber was perfused with a BRB80-based motility buffer supplemented with 1 mM ATP, 25 mM KCl, 25 μM taxol, 1.3 mg ml$^{-1}$ casein, and an oxygen scavenger system[57]. Time-lapse images were acquired at 1 frame per 5 s for 3 min.

For microtubule gliding assay, the chamber was perfused with anti-His antibody diluted in BRB80 and incubated for 2 min at room temperature. After washing away unbound antibody, one chamber of motor properly diluted in BRB80 supplemented with 20 μM taxol and 1.3 mg ml$^{-1}$ casein was added to the chamber. After 2 min incubation at room temperature, unbound motors were washed away with BRB80 supplemented with 20 μM taxol and 1.3 mg ml$^{-1}$ casein. Polarity-marked TMR microtubule diluted in BRB80 supplemented with 20 μM taxol and 1.3 mg ml$^{-1}$ casein was then added to the chamber. After 2-min incubation, unbound microtubules were removed by washing the chamber with two chamber volumes of BRB80 supplemented with 20 μM taxol and 1.3 mg ml$^{-1}$ casein. Finally, the flow chamber was perfused with one chamber volume of BRB80-based motility buffer containing 1 mM ATP, 25 mM KCl, 25 μM taxol, 1.3 mg ml$^{-1}$ casein, and an oxygen scavenger system. Time-lapse images were taken at 1 frame per 5 s for 5 min.

For the single-molecule motility assays, taxol-stabilized polarity-marked HiLyte 647/Biotin-labeled microtubules were first immobilized using the same procedure as described above in the actin transport assay. After removing unbound microtubules by washing the chamber with five chamber volumes of BRB15 supplemented with 20 μM taxol and 1.3 mg ml$^{-1}$ casein, the chamber was perfused with a BRB80-based motility mixture containing diluted KCH motors, 1 mM ATP, 25 mM KCl, 25 μM taxol, 1.3 mg ml$^{-1}$ casein, and an oxygen scavenger system as previously described[57]. Time-lapse image sequences of the processive GFP-OsKCH2(289–767) were recorded at 1 frame per 2 s with an exposure time of 200 ms for 10 min. Kymographs were generated and analyzed in ImageJ (NIH) for extracting the velocity and run-length information of individual KCH motors. Only those kymograph lines containing at least 4 pixels on both the X- and Y-axes in the kymographs were included in the analysis. Time-lapse image sequences of the nonprocessive GFP-OsKCH2(289–720), GFP-OsKCH1(292–744), and GFP-K760A/R761A/R764A/R766A were recorded continuously with an exposure time of 200 ms for 1 min.

For the single-molecule photobleaching assays, experiments were performed in the single-molecule motility buffer without ATP to keep the kinesin molecules tightly bound to surface-immobilized microtubules. Time-lapse image sequences

were continuously recorded with an exposure time of 100 ms until the field of view was completely bleached of fluorescence signal. The number of photobleaching steps of individual kinesin motors was obtained by measuring the fluorescence intensity in ImageJ.

**Sucrose gradient centrifugation.** To determine the oligomer status of a KCH motor, 50–100 μl of purified protein was loaded on the top of a 5–20% sucrose gradient prepared in BRB80 supplemented with 25 mM KCl as previously described[58]. The gradient was centrifuged at $150,000 \times g$ at 4 °C for 18 h in the Beckman SW40 rotor. Fractions (~450 μl) were collected from the top and analyzed by the standard SDS-PAGE. The peak fraction was determined by fitting the protein band intensities with a Gaussian distribution. The sedimentation value was determined using the following standards: carbonic anhydrase (CA), MW = 29 kDa, 2.8 S; BSA, MW = 65 kDa, 4.4 S; alcohol dehydrogenase (AD), MW = 150 kDa, 7.4 S; amylase, MW = 200 kDa, 8.98 S; and apoferritin, MW = 440 kDa, 17.2 S.

**Microtubule co-sedimentation assays.** Microtubule co-sedimentation assays were performed as previously described[59]. Briefly, taxol-stabilized microtubules were polymerized from unlabeled tubulin (200 μM) using the aforementioned protocol. Purified KCH constructs—GST-OsKCH2(721–767), GFP-OsKCH2 (289–767), GFP-OsKCH2(289–720), GFP-K760A/R761A/R764A/R766A, GFP-OsKCH1(292–744), and GFP-OsKCH1/OsKCH2—were each mixed with micro-tubules in a BRB80 buffer supplemented with 40 μM taxol, incubated at room temperature for 30 min, and centrifuged at $100,000 \times g$ using a TLA100 rotor (Beckman Coulter) for 20 min at 37 °C. Coomassie-stained SDS-PAGE gels were analyzed to compare the protein amount in the supernatant and pellet fractions.

**Data availability.** The data that support the findings of this study are available from the corresponding author on reasonable request.

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

## Acknowledgements

We thank C. Larson and R. Scheirer for valuable initial work; Drs. X. Su (UCSF), and C. K. Mathews and P.A. Karplus (Oregon State University) for critical reading of the manuscript; and members of the Qiu lab for helpful discussions. P.W. was supported by a visiting student scholarship from the China Scholarship Council and through grants from the National Science Foundation Committee of China (Project No. U1604129 and 21173068 to L.G.). This work is supported by the NSF grants MCB-1616076 (to B.L.) and MCB-1616462 (to W.Q.).

## Author contributions

B.L. and W.Q. conceived, designed, and supervised the study; Y.-R.J.L. and B.L. discovered OsKCH2, isolated its cDNA, and performed the immunolocalization experiments; K.-F.T., P.W., J.B., A.M.G., and L.G. performed all other experiments and analyses; W.Q. wrote the manuscript with input from all authors.

## Additional information

**Competing interests:** The authors declare no competing interests.

