## [Peer Review File · Nature Communications]

Reviewers' comments:

Reviewer #1 (Remarks to the Author):

This manuscript reports motility of a rice kinesin-14, OsKCH2, which contains an actin-binding calponin homology domain. The authors show that OsKCH2 is processive and identify a region of the motor that is required for processivity and apparently acts by binding the motor to MTs.

The findings reported here are not novel and represent only a small advance in available information for the kinesin-14 motors: processive movement by OsKCH2 differs from most, but not all kinesin-14s (Ref 17-19); other plant kinesins have been reported to bind both to actin and MTs (Ref 27-32); and another rice kinesin-14, OsKCH1, has previously been reported to transport actin filaments along MTs (Ref 16). My enthusiasm regarding this manuscript is therefore not high.

The following issues also need to be addressed:

1. The relationship between OsKCH2 and OsKCH1 is not given. Phylogenetic analysis should be performed to show the relationship between the two proteins and among the known rice kinesin-14s, as well as their relationship to kinesin-14s from another plant species, e.g., Arabidopsis. This analysis would address the questions of whether OsKCH2 and OsKCH1 are close homologues with divergent mechanisms of function and whether there are other rice kinesin-14s that are expected to show functional redundancy with OsKCH2.
2. CC2 is reported here as a coiled-coil domain that does not appear to dimerize OsKCH2. What is the basis of its identification as a coiled-coil domain? Coiled coils have a characteristic amino acid signature - the analysis of this domain, along with that of CC1, should be presented, not just shown with charged amino acids highlighted as in Sup Fig 2.
3. CC2 is reported to be essential for OsKCH2 processivity, but it binds very weakly to MTs under (nonphysiological) salt conditions when expressed alone, although it enhances OsKCH2 binding to MTs. The authors should provide further experiments that explain the role of CC2 in OsKCH2 processivity.
4. How does the processivity of OsKCH2 relate to its cellular function? The authors do not present any information regarding the cellular localization or expression of OsKCH2 that bears on its motility - this information is needed to interpret the relevance of processivity to OsKCH2 function in cells.

Reviewer #2 (Remarks to the Author):

The manuscript entitled "OsKCH2 is a processive minus end directed kinesin-14 motor" is a detailed characterization of the biophysical properties of a plant specific kinesin-14 member. The experiments are solid, well presented and the detailed materials and methods and references therein appear appropriate.

The key finding here is the somewhat unexpected processivity of the rice OsKCH2 kinesin-14 minus-end directed motor protein. As the authors point out it is the first report of a processive plant kinesin-14, which could potentially compensate for the lack of cytoplasmic dynein from the plant lineage. This finding is not only relevant for plant cell biology, but provides insight for a general understanding of how the super-family of kinesin motors diversified their functions. Apparently, the processive behavior of OsKCH2 relies on the presence of a positively charged coiled coil domain, CC2, located towards the C-terminus. Processivity is abolished in the absence of the domain CC2. Furthermore, CC2 alone does

co-sediment with microtubules in co-sedimentation assays.

I have few comments and suggestions:

The photobleaching experiment suggests that OsKCH2 (and OsKCH1 for that matter) act as homodimers. However, in Figure 3b which shows the frequency of bleaching steps; here also 3-steps and 4-steps are counted. The sucrose gradient however supports the claim that OsKCH2 act as homodimers. Can the authors address the discrepancy?

Below follows a short list of experiments for discussion. Each thereof, would significantly reinforced the conclusion that CC2 is responsible for processivity, in my opinion.

1. Mutation of a number of positive charges to alanine, in the context of GFP-OsKCH2(289-767) should reduce/abolish processivity.
2. Grafting of OsKCH1 CC2 onto OsKCH2(289-720) should turn OsKCH1 into a processive motor. Alternatively, the reciprocal grafting of the OsKCH1 CC2 onto OsKCH2(289-720) could be done, expecting loss of processivity, but loss of activity is not so informative.
3. Co-sedimentation assay of OsKCH1 CC2, although this might not be as insightful, if it co-sediments with microtubules to the same extent as OsKCH2 does.

Reviewer #3 (Remarks to the Author):

Review Tseng et al. Nature 2017

The manuscript describes experiments to characterize the biochemical properties of a plant kinesin 14 isolated from rice. The major claim is that the specific kinesin shows processive motility toward the minus end of microtubules as a dimer. The importance of the discovery is explained in the broader context of land plant evolution where the minus end directed motor, cytoplasmic dynein, is not present to affect a host of important cellular phenomena.

After reading this manuscript through multiple times, I will save the typical sniping and just say that it is, in my opinion, not at a level of general interest for Nature. The narrow finding of a processive, minus end directed motor in plants is not the same as finding a motor with a demonstrated function to effectively 'replace' dynein function(s), if such activity is truly required in these acentrosomal systems. With that said, I have to comment that this is an extremely well prepared and well developed study with an important central discovery about the domains in this, and the related, kinesin 14 motors. I have several minor comments that I hope will be of help to the authors in their exploration of this motor, beyond the biochemistry.

The authors provide what is effectively a negative result to show that the CC2 domain is critical for the processivity in the KCH2 protein. That experiment requires a fair bit of scrutiny where the full length protein is not tested under various presumptions about the c-terminal function etc. Based on the homology claimed between the non-processive KCH1 and KCH2, what happens in the domain swap with the KCH1 protein? Does the CC2 domain confer processivity in that experiment to KCH1? I understand from the discussion section that the authors are attempting to get structural data to infer attributions to the domains in question. But the mechanism leading to the somewhat novel shift to a processive activity, claimed in this work, may not be immediately apparent from the structural data. A functional determination of CC2 domain action could go a long way toward showing why this domain can lead to the extended walk of the dimer on the polymer.

Does the KCH2 mutant have phenotypes that would make the broader narrative about dynein 'replacement' more direct? Without going through the list of activities, the more visible roles for

dynein in both interphase and mitosis seem to be less apparent or even absent in plant cell biology. Is there a specific phenotype that might lend itself to the larger understanding of why plants still need processive minus end motors?

As a technical point, the claim made in section 1 about the motor having 2 distinct velocities does not seem completely merited or, at least, requires some clarification. The kymograph in 1e shows a faster and a slower movement pointed out with arrows. But the kymograph also shows a motor either paused and moving to a faster velocity –or a motor shifting from slower to faster. The histogram in 1f shows a skewed distribution, but that designation of the peaks from the multi-Gaussian fitting model is not terribly convincing. Are the individual motor movements bi-modal (either fast or slow) or are the motors changing velocity over the course of their processive runs? What is the mean fluctuation in the motor velocity for each run, plotted against the mean velocity (expected linear scaling of error, or is it skewed like the velocity histogram)? Noting that the effort was made to correlate the bleaching events with the motor stoichiometry (indicating reproducible relative intensity measures), is there a relationship between the mean fluorescence intensity and velocity that could indicate higher order complexes? And finally, how do the velocities of each run correspond to the processivity; if you plotted mean velocity against run length, is scaling for distance with velocity, or is it random? The issue and justification for why this is potentially bi-modal is not cleared up in the manuscript and not explained in light of the unimodal microtubule gliding assays.

As a final comment, I want to commend the authors for the use of the diagrams and the clarity of the figures in most cases. The paper is well constructed and clearly laid out.

AUTHOR RESPONSE:

We thank all reviewers for their comments and suggestions, which have helped us to substantially improve our manuscript in this revision. We hope that it will now be acceptable for publication in *Nature Communications*. In the following we address, point-by-point, each of the reviewers' comments, with the reviewers' comments in blue and our responses in plain text. Please note that changes in the manuscript are highlighted in blue.

Reviewer #1 (Remarks to the Author):

This manuscript reports motility of a rice kinesin-14, OsKCH2, which contains an actin-binding calponin homology domain. The authors show that OsKCH2 is processive and identify a region of the motor that is required for processivity and apparently acts by binding the motor to MTs.

The findings reported here are not novel and represent only a small advance in available information for the kinesin-14 motors: processive movement by OsKCH2 differs from most, but not all kinesin-14s (Ref 17-19); other plant kinesins have been reported to bind both to actin and MTs (Ref 27-32); and another rice kinesin-14, OsKCH1, has previously been reported to transport actin filaments along MTs (Ref 16). My enthusiasm regarding this manuscript is therefore not high.

While we agree with the reviewer that other plant kinesins have been shown to crosslink actin filaments and microtubules and to transport actin filaments along microtubules, we emphasize that OsKCH2 is in fact the only kinesin-14 studied so far that shows processive minus-end-directed motility on single microtubules as a homodimer. Specifically, the processive motility of OsKCH2 is quite distinct from the other two processive kinesin-14s Kar3 and KlpA, because the processive minus-end-directed motility of Kar3 on single microtubules requires a *heterodimer* and the processive motility that KlpA achieves as a homodimer is *plus-end-directed*. As the reviewer will see below in one of our responses, we have acquired and included in this revision additional localization data to show that OsKCH2 localizes to the microtubule bundles in the preprophase band. Thus, as pointed out by Reviewer #2, our study “is not only relevant for plant cell biology, but provides insight for a general understanding of how the super-family of kinesin motors diversified their functions.”

The following issues also need to be addressed:

1. The relationship between OsKCH2 and OsKCH1 is not given. Phylogenetic analysis should be performed to show the relationship between the two proteins and among the known rice kinesin-14s, as well as their relationship to kinesin-14s from another plant species, e.g., *Arabidopsis*. This analysis would address the questions of whether OsKCH2 and OsKCH1 are close homologues with divergent mechanisms of function and whether there are other rice kinesin-14s that are expected to show functional redundancy with OsKCH2.

KCHs represent a special group within the extensively expanded kinesin-14 subfamily in flowering plants. Based on the reviewer comments, we have performed a phylogenetic analysis of all kinesin-14 proteins from *Arabidopsis thaliana* and *Oryza sativa*. As illustrated in Supplementary Figure 2, KCHs from *O. Sativa* and *A. thaliana* form a branch made of two primary clades. In the top clade, OsKCH2 is more closely related to the KCH protein AtKP1 from *A. thaliana* than to other three rice KCHs including OsKCH1. We have revised the manuscript to include the phylogenetic analysis result, and further suggest in the discussion that future studies are also needed to investigate whether AtKP1 retained inherent minus-end-directed processive microtubule-base motility after the speciation process.

2. CC2 is reported here as a coiled-coil domain that does not appear to dimerize OsKCH2. What is the basis of its identification as a coiled-coil domain? Coiled coils have a characteristic amino acid signature - the analysis of this domain, along with that of CC1, should be presented, not just shown with charged amino acids highlighted as in Sup Fig 2.

All coiled-coil analyses in our study were performed using the program MARCOIL(Delorenzi and Speed, 2002). Based on the reviewer's suggestion, we have shown the coiled-coil profile of OsKCH2 in Supplementary Figure 1 and that of OsKCH1 in Supplementary Figure 4a. Two things are worthy of attention.

- (1) We previously mistakenly stated that the upstream coiled-coil CC1 covers aa 289-359, and this has been corrected in this revision (CC1, aa 313-359).
- (2) In this revision, we use a cutoff probability of 80% for the final coiled-coil assignment and report three predicted coiled-coils in OsKCH2: two upstream coiled-coils (CC0, aa 239-295; CC1, aa 313-354) and one downstream coiled-coil (CC2, aa 718-766). In our previous submission, a more stringent cutoff probability of 90% was used, which led to the identification of a single upstream coiled-coil (CC1) and a downstream coiled-coil (CC2).

We have revised the manuscript and the figures to reflect these changes. We emphasize that these changes do not affect the conclusions of our manuscript. Nevertheless, we apologize for any confusion these changes may incur. We thank the reviewer for the comments, which allowed us to identify and correct the mistake concerning CC1.

3. CC2 is reported to be essential for OsKCH2 processivity, but it binds very weakly to MTs under (nonphysiological) salt conditions when expressed alone, although it enhances OsKCH2 binding to MTs. The authors should provide further experiments that explain the role of CC2 in OsKCH2 processivity.

We have performed the following experiments to further clarify the role of CC2 in OsKCH2 processivity.

- (1) We engineered the nonprocessive GFP-OsKCH1(292-744) to replace its endogenous CC2 with that of OsKCH2, and our single-molecule motility experiments showed that the resulting GFP-OsKCH1/KCH2 chimera becomes a processive minus-end-directed motor on single microtubules (Fig. 5a-e). In addition, microtubule co-sedimentation experiments showed that the processive GFP-OsKCH1/KCH2 chimera binds to the microtubule much more tightly than the nonprocessive GFP-OsKCH1(292-744) (Fig. 5f,g). Thus, despite the relatively weak microtubule binding we saw for CC2 from OsKCH2 alone, in the context of the dimeric motor construct it quite definitely enhances the microtubule binding and is sufficient to enable OsKCH1 for processive motility on single microtubules.
- (2) We created a quadruple mutant of the processive GFP-OsKCH2(289-767) to replace the last four positively charged residues in CC2 to the neutral alanine (Supplementary Fig. 6a). We found that the resulting quadruple mutant (GFP-K760A/R761A/R764A/R766A) retained the ability to collectively glide microtubules with minus-end-directed motility (Supplementary Fig. 6b), but behaved like a nonprocessive motor on single microtubules (Supplementary Fig. 6c) and showed significantly reduced microtubule-binding affinity (Supplementary Fig. 6d). This suggests that these charged residues in CC2 are important for the processivity of OsKCH2.

These additional data further support our hypothesis that CC2 in OsKCH2 enables the motor for processive motility on single microtubules by acting as an ATP-independent microtubule-binding site.

4. How does the processivity of OsKCH2 relate to its cellular function? The authors do not present any information regarding the cellular localization or expression of OsKCH2 that bears on its motility - this information is needed to interpret the relevance of processivity to OsKCH2 function in cells.

At present, precisely defining the physiological function(s) of OsKCH2 and the implication of its intrinsic processivity is compounded by the lack of an aberrant phenotype associated with the homozygous *kch2* mutant (we suspect due to redundancy of OsKCH2 with at least one of the other 8 KCH kinesins in rice). Nonetheless, we have now successfully generated specific antibodies against OsKCH2, and obtained and included in this revision additional immunolocalization data showing that OsKCH2 forms a punctate localization pattern along the PPB microtubules at prophase (Fig. 1b). Based on previous reports that actin filaments (AFs) transiently associate with the PPB microtubule bundles (Ding et al., 1991; Kakimoto and Shibaoka, 1987; Liu and Palevitz, 1992; Palevitz, 1987) and our finding that OsKCH2 is able to transport AFs on the microtubules with minus-end-directed motility *in vitro*, we suggest “one likely function of OsKCH2 is to recruit AFs to the PPB by either dynamically translocating AFs on the PPB microtubules or statically crosslinking these two cytoskeletal filaments inside the PPB”. We agree that more physiological studies are needed, but they are beyond the scope of this work. We anticipate that our study – which is the first characterization of OsKCH2 – will stimulate more genetic studies and other analyses to reveal the *in vivo* function(s) of OsKCH2 and the role of its processive motility.

Reviewer #2 (Remarks to the Author):

The manuscript entitled “OsKCH2 is a processive minus end directed kinesin-14 motor” is a detailed characterization of the biophysical properties of a plant specific kinesin-14 member. The experiments are solid, well presented and the detailed materials and methods and references therein appear appropriate.

The key finding here is the somewhat unexpected processivity of the rice OsKCH2 kinesin-14 minus-end directed motor protein. As the authors point out it is the first report of a processive plant kinesin-14, which could potentially compensate for the lack of cytoplasmic dynein from the plant lineage. This finding is not only relevant for plant cell biology, but provides insight for a general understanding of how the super-family of kinesin motors diversified their functions. Apparently, the processive behavior of OsKCH2 relies on the presence of a positively charged coiled coil domain, CC2, located towards the C-terminus. Processivity is abolished in the absence of the domain CC2. Furthermore, CC2 alone does co-sediment with microtubules in co-sedimentation assays.

We thank the reviewer for the enthusiasm and the strong support.

I have few comments and suggestions:

The photobleaching experiment suggests that OsKCH2 (and OsKCH1 for that matter) act as homodimers. However, in Figure 3b, which shows the frequency of bleaching steps; here also 3-steps and 4-steps are counted. The sucrose gradient however supports the claim that OsKCH2 act as homodimers. Can the authors address the discrepancy?

Single-molecule photobleaching is a standard assay for determining kinesin oligomerization. As noted by the reviewer, photobleaching histograms of GFP-OsKCH2(289-767) and GFP-OsKCH1(292-744) both contain 3- and 4-step events. Others have similarly observed 3- and 4-step events in their photobleaching experiments of GFP-tagged dimeric kinesins, such as FRA1 from *Arabidopsis thaliana* (see Fig. 4b in Zhu and Dixit, 2011) and Kinesin14-VIb from *Physcomitrella patens* (see Fig. 2d in Jonsson et al., 2015). While we could not rule out the possibility that some of those 3- and 4-steps are due to nonspecific formation of high-order oligomers consisting of more than two OsKCH2 (or OsKCH1) monomers, we believe that the majority of those 3- and 4-step events are caused by inadvertent co-localization of two

OsKCH2 (or OsKCH1) dimers. Strong supporting evidence comes from the single-molecule motility experiments of GFP-OsKCH1(282-744): nonprocessive kinesin-14 motors are known to achieve processive motility by clustering to contain two dimers(Furuta et al., 2013; Jonsson et al., 2015), but in spite of the relatively higher percentage of 3- and 4-steps (~28%) (Supplementary Fig. 4e) no processive motility was observed for GFP-OsKCH1(292-744) (Supplementary Fig. 4g). Furthermore, regardless of the exact sources of these 3- and 4-step events, all photobleaching histograms in this study are dominated by 1- and 2-step events, suggesting that dimeric kinesins are the main species in our experimental conditions for GFP-OsKCH2(289-767), GFP-OsKCH1(292-744) and GFP-OsKCH1/OsKCH2. For GFP-OsKCH2(289-767), the photobleaching histogram (Fig. 3b) and the sucrose gradient result (Fig. 3c) are consistent with each other and both show that GFP-OsKCH2(289-767) exists predominantly as a homodimer in our experimental condition. It is also worth mentioning that 1-step events are mostly due to prebleaching of one GFP before the start of counting but could also be due to incomplete maturation of some GFPs.

Below follows a short list of experiments for discussion. Each thereof, would significantly reinforced the conclusion that CC2 is responsible for processivity, in my opinion.

1. Mutation of a number of positive charges to alanine, in the context of GFP-OsKCH2(289-767) should reduce/abolish processivity.

Yes, the processive GFP-OsKCH2(289-767) can be rendered nonprocessive by mutating some positively charged residues in CC2 to the neutral alanine. Please see more details in our response to Comment #3 from Reviewer #1.

2. Grafting of OsKCH1 CC2 onto OsKCH2(289-720) should turn OsKCH1 into a processive motor. Alternatively, the reciprocal grafting of the OsKCH1 CC2 onto OsKCH2(289-720) could be done, expecting loss of processivity, but loss of activity is not so informative.

We think the reviewer meant to suggest grafting the CC2 of OsKCH2 onto OsKCH1. Based on this suggestion, we have engineered a chimeric construct GFP-OsKCH1/KCH2 from the nonprocessive GFP-OsKCH1(292-744) by substituting its endogenous CC2 with that of the processive GFP-OsKCH2(289-767) (Fig. 5a). Our results show that the GFP-OsKCH1/KCH2 chimera indeed moves processively on single microtubules as a homodimer. Please see more details in our response to Comments #3 from Reviewer #1.

Based on the results of the chimera GFP-OsKCH1/KCH2 and the quadruple mutant GFP-K760A/R761A/R764A/R766A, we decided not to pursue the experiment to graft CC2 of OsKCH1 onto the CC2-less GFP-OsKCH2(289-720) to generate the chimera GFP-OsKCH2/KCH1. We agree with the reviewer that it is highly unlikely GFP-OsKCH2/KCH1 will be able to exhibit processive motility on single microtubules as a homodimer.

3. Co-sedimentation assay of OsKCH1 CC2, although this might not be as insightful, if it co-sediments with microtubules to the same extent as OsKCH2 does.

We have performed the microtubule co-sedimentation for CC2 of OsKCH1. Our results showed that CC2 of OsKCH1 barely binds to microtubules (Author Response Fig. 1). We agree with the reviewer that this result is not as insightful as those of the quadruple mutant GFP-K760A/R761A/R764A/R766A and the chimera GFP-GFP-OsKCH1/KCH2, and thus have chosen not to include this result in the revision.

Author Response Figure 1. CC2 of OsKCH1 does not exhibit obvious microtubule binding. **a**, Schematic of the full-length OsKCH1 and GST-OsKCH1(704-744). **b**, Coomassie-stained SDS-PAGE of the microtubule co-sedimentation assay for GST-OsKCH1(706-744) in BRB80/25 mM KCl. **c**, Coomassie-stained SDS-PAGE of the microtubule co-sedimentation assay for GST-OsKCH1(706-744) in the low ionic strength solution BRB12.

Reviewer #3 (Remarks to the Author):

Review Tseng et al. Nature 2017

The manuscript describes experiments to characterize the biochemical properties of a plant kinesin 14 isolated from rice. The major claim is that the specific kinesin shows processive motility toward the minus end of microtubules as a dimer. The importance of the discovery is explained in the broader context of land plant evolution where the minus end directed motor, cytoplasmic dynein, is not present to affect a host of important cellular phenomena.

After reading this manuscript through multiple times, I will save the typical sniping and just say that it is, in my opinion, not at a level of general interest for Nature. The narrow finding of a processive, minus end directed motor in plants is not the same as finding a motor with a demonstrated function to effectively ‘replace’ dynein function(s), if such activity is truly required in these acentrosomal systems. With that said, I have to comment that this is an extremely well prepared and well developed study with an important central discovery about the domains in this, and the related, kinesin 14 motors. I have several minor comments that I hope will be of help to the authors in their exploration of this motor, beyond the biochemistry.

We are glad that the reviewer found our work “is an extremely well prepared and well developed study with an important central discovery”. We agree with the reviewer that additional studies – in particular studies that could conclusively demonstrate OsKCH2 is a functional replacement of dynein in rice and requires intrinsic processive minus-end-directed motility to achieve that feat – would further boost the impact and degree of general interest of our study. However, we are very pleased that the localization studies we added in response to Reviewer #1 add further significance to our work by conclusively showing that this protein is involved in a very important plant-specific physiological process.

The authors provide what is effectively a negative result to show that the CC2 domain is critical for the processivity in the KCH2 protein. That experiment requires a fair bit of scrutiny where the full length protein is not tested under various presumptions about the c-terminal function etc. Based on the homology claimed between the non-processive KCH1 and KCH2, what happens in the domain swap with the KCH1 protein? Does the CC2 domain confer processivity in that experiment to KCH1? I understand from the discussion section that the authors are attempting to get structural data to infer attributions to the domains in question. But the mechanism leading to the somewhat novel shift to a processive activity, claimed in this work, may not be immediately apparent from the structural data. A functional determination of CC2 domain action could go a long way toward showing why this domain can lead to the extended walk of the dimer on the polymer.

We have done the domain-swapping experiment to show that CC2 from OsKCH2 enables OsKCH1 for processive motility on single microtubules. Please see more details in our response to Comments #3 from Reviewer #1.

Does the KCH2 mutant have phenotypes that would make the broader narrative about dynein 'replacement' more direct? Without going through the list of activities, the more visible roles for dynein in both interphase and mitosis seem to be less apparent or even absent in plant cell biology. Is there a specific phenotype that might lend itself to the larger understanding of why plants still need processive minus end motors?

By screening mutants carrying out TOS17 transposon insertions (Miyao, 2003), we have isolated a homozygous *kch2* mutant. The homozygous *kch2* mutant cells lacked OsKCH2 at the PPB when examined by anti-OsKCH2 immunofluorescence (unpublished data). However, as noted above, the homozygous *kch2* mutant did not exhibit any noticeable phenotype, which is thought to be due to functional redundancy among two or more of the 9 KCH motors in rice. Thus, more extensive genetic analyses are needed in order to reveal the *in vivo* function(s) of OsKCH2 and the role of its processive motility once appropriate mutants lacking more than 1 KCH are identified in the future.

As a technical point, the claim made in section 1 about the motor having 2 distinct velocities does not seem completely merited or, at least, requires some clarification. The kymograph in 1e shows a faster and a slower movement pointed out with arrows. But the kymograph also shows a motor either paused and moving to a faster velocity – or a motor shifting from slower to faster. The histogram in 1f shows a skewed distribution, but that designation of the peaks from the multi-Gaussian fitting model is not terribly convincing. Are the individual motor movements bi-modal (either fast or slow) or are the motors changing velocity over the course of their processive runs? What is the mean fluctuation in the motor velocity for each run, plotted against the mean velocity (expected linear scaling of error, or is it skewed like the velocity histogram)? Noting that the effort was made to correlate the bleaching events with the motor stoichiometry (indicating reproducible relative intensity measures), is there a relationship between the mean fluorescence intensity and velocity that could indicate higher order complexes? And finally, how do the velocities of each run correspond to the processivity; if you plotted mean velocity against run length, is scaling for distance with velocity, or is it random? The issue and justification for why this is potentially bi-modal is not cleared up in the manuscript and not explained in light of the unimodal microtubule gliding assays.

We agree that more quantitative analyses are needed to clarify the bimodal velocity distribution of actin filament transport by OsKCH2(1-767) along microtubules *in vitro*, but we wish to point out that the main conclusions of our current work does not hinge on the precise velocity distribution of actin transport by OsKCH2(1-767). The main purpose of the *in vitro* actin transport experiments was to show that OsKCH2 not only simultaneously interacts with actin filaments and microtubules but also actively transports actin filaments along microtubules *in vitro*. The analyses suggested by the reviewer would be an important next

step to further characterize the biophysics of actin filament transport by OsKCH2(1-767) along microtubules *in vitro*. However, these analyses will be a substantial undertaking and are thus beyond the scope of this study.

In addition, Walter and colleagues have previously investigated in great detail the transport of actin filament by the nonprocessive OsKCH1 along microtubules *in vitro* (Walter et al., 2015). In that study, the authors found that OsKCH1 clusters to transport actin filaments along microtubules with two distinct velocities, and proposed that the actin transport velocity is dependent on the orientation of an actin filament on the microtubule. In this revision, we have directed the attention of interested readers to the Walter et al study.

As a final comment, I want to commend the authors for the use of the diagrams and the clarity of the figures in most cases. The paper is well constructed and clearly laid out.

We thank the reviewer for the commendation.

Reference:

Delorenzi, M., and Speed, T. (2002). An HMM model for coiled-coil domains and a comparison with PSSM-based predictions. *Bioinformatics* 18, 617–625.

Ding, B., Turgeon, R., and Parthasarathy, M.V. (1991). Microfilaments in the preprophase band of freeze substituted tobacco root cells. *Protoplasma* 165, 209–211.

Furuta, K., Furuta, A., Toyoshima, Y.Y., Amino, M., Oiwa, K., and Kojima, H. (2013). Measuring collective transport by defined numbers of processive and nonprocessive kinesin motors. *Proc. Natl. Acad. Sci. U.S.a.* 110, 501–506.

Jonsson, E., Yamada, M., Vale, R.D., and Goshima, G. (2015). Clustering of a kinesin-14 motor enables processive retrograde microtubule-based transport in plants. *Nat. Plants* 1, 15087.

Kakimoto, T., and Shibaoka, H. (1987). Actin filaments and microtubules in the preprophase band and phragmoplast of tobacco cells. *Protoplasma* 140, 151–156.

Liu, B., and Palevitz, B.A. (1992). Organization of cortical microfilaments in dividing root cells. *Cytoskeleton* 23, 252–264.

Miyao, A. (2003). Target site specificity of the Tos17 retrotransposon shows a preference for insertion within genes and against insertion in retrotransposon-rich regions of the genome. *Plant Cell* 15, 1771–1780.

Palevitz, B.A. (1987). Actin in the preprophase band of *Allium cepa*. *J. Cell Biol.* 104, 1515–1519.

Walter, W.J., Machens, I., Rafeian, F., and Diez, S. (2015). The non-processive rice kinesin-14 OsKCH1 transports actin filaments along microtubules with two distinct velocities. *Nat. Plants* 1, 15111.

Zhu, C., and Dixit, R. (2011). Single molecule analysis of the Arabidopsis FRA1 kinesin shows that it is a functional motor protein with unusually high processivity. *Mol. Plant* 4, 879–885.

Reviewers' comments:

Reviewer #1 (Remarks to the Author):

The authors have provided some additional data in this revised manuscript, but there are basic problems with this ms: there are no controls for some of the additional data and the added data do not dispel my original assessment that the findings reported here are not novel. I therefore do not recommend that this manuscript be accepted for publication by Nature Communications.

The processive motility of the rice kinesin-14 OskCH2 motor is not novel, as noted previously, nor is its interaction with both MTs and actin filaments. The fact that basic residues increase MT binding and can convert a nonprocessive motor into a processive motor was demonstrated some time ago for a kinesin motor. The results reported here are therefore not novel. They also lack biological significance, given that no function has been found for OskCH2 - the immunolocalization data presented here lack essential controls and there is no loss-of-function phenotype that has been identified.

There are essential missing controls and other problems with this revised ms:

1. The authors have now decided that there are two upstream putative coiled coils, CC0 and CC1, not just one, although they continue to refer to CC1 as if it is the only upstream coiled coil. The coiled coil analysis they present is highly confusing, as it lacks positive and negative controls. The authors need to compare their program predictions with those for a known coiled coil, established using structural methods, and also with a protein whose known structure does not include a coiled coil to determine the validity of their coiled coil assignments based on the program predictions alone.

It is not clear whether CC0 is a distinct coiled coil from CC1 - it appears to be the same coiled coil as CC1 with a hinge region. Moreover, CC2 in OskCH2 has some of the features of a coiled coil, but the heptad repeat motif appears to deviate from a coiled coil in Sup Fig S5: a coiled coil is characterized by the hydrophobic residues in the two strands interacting with each other and it is not certain that this is the case for the region denoted as CC2. The region also does not appear to be a coiled coil in OskCH1 by examination of the protein sequence. Note that the program should find the heptad repeat regions but the characteristics of the regions need to be analyzed further to determine whether the number of heptad repeats and the characteristics of the residues are those that are found in known coiled coils. From Fig S5, the authors may be incorrect in referring to this region as CC2. Their data indicating that the region does not dimerize the protein argues against the residues in this region adopting a coiled coil conformation, even if the predictive program they are using identifies it as such.

They also should mention that CC0 is missing in the 289-767 construct - this should be made clear throughout the ms. What is the role of CC0?

2. The phylogenetic analysis also lacks controls and essential information. The alignment used for the tree building is not shown and should be presented. Did the alignment include only the motor domain sequences and was it optimized? Did the tree search include an outgroup protein to determine whether all the proteins shown in the final tree grouped together? Were controls performed in which the protein sequences in the alignment were randomized and the tree search redone? Were other methods used to validate the groupings shown, e.g., pairwise comparisons of sequences to show that the proteins that group together are more closely related to one another than those in other groups? The protein names shown for the tree should be made shorter and the abbreviations and other information given in the figure legend to make the groupings in the tree easier to see at a glance.

3. The antibodies against OskCH2 are described as "monospecific" but there are no data showing that

the antibodies recognize a specific epitope in OsKCH2. There are also no control experiments showing that the antibodies are specific for OsKCH2 and do not cross react with OsKCH1, another rice kinesin-14, or another protein. There are also no controls showing that the immunofluorescence is localized to the cell cortex surrounding the prophase nucleus - co-staining by a cortical antibody is needed to demonstrate this. The authors should also show the negative mutant cells, alongside the missing positive controls. If the authors want to conclude that the localization is to the preprophase band, they need to show a positive control that shows specific antibody staining to the preprophase band. The OsKCH2 staining in Fig 1b does not resemble the diagram of the PPB in Fig 1b. The biological significance of the OsKCH2 motor processivity is not at all clear, especially if it is redundant with one or more of the other rice KCH kinesins, as the authors suggest in their rebuttal as the reason for not obtaining a mutant phenotype.

4. The bimodality of the velocity in Fig 1g is not apparent from the histogram, as also noted by another reviewer. What is the rationale for assigning two distinct velocities to the motor (other than the previous report for OsKCH1)? Why are these velocities for actin filament transport so much slower than the minus-end movement of the 289-767 OsKCH2 motor?

5. The authors should note that although their sedimentation data are consistent with OsKCH2 being a homodimer in solution, these data and their photobleaching assays do not rule out the possibility that two homodimers couple in the motility assays. Note that the photobleaching assays were performed under different conditions and geometry than the motility assays. They state that they have demonstrated processive motility without clustering but they have not shown that the processive motility they observe is **not** due to clustering.

Reviewer #2 (Remarks to the Author):

In the revised manuscript, the authors further investigated the origin of processivity in OsKCH2 by analysis of chimera between OsKCH2 and OsKCH1 and by mutation of a stretch of basic residues in CC2, expected to modulate the interaction between the kinesin and microtubules. Furthermore, the authors include *in vivo* co-localization analysis of OsKCH2 with the preprophase band and extend the manuscript by a phylogenetic tree and extend the protein domain analysis. In my opinion, these revisions adequately addressed the major concerns raised by the reviewers.

Response to reviewers' comments:

In the following, we will address, point-by-point, each of the comments from the reviewers, with the reviewer comments in blue and our responses in plain text. Please note that changes in the manuscript are highlighted in blue.

Reviewer #1 (Remarks to the Author):

The authors have provided some additional data in this revised manuscript, but there are basic problems with this ms: there are no controls for some of the additional data and the added data do not dispel my original assessment that the findings reported here are not novel. I therefore do not recommend that this manuscript be accepted for publication by Nature Communications.

The processive motility of the rice kinesin-14 OsKCH2 motor is not novel, as noted previously, nor is its interaction with both MTs and actin filaments. The fact that basic residues increase MT binding and can convert a nonprocessive motor into a processive motor was demonstrated some time ago for a kinesin motor. The results reported here are therefore not novel. They also lack biological significance, given that no function has been found for OsKCH2 - the immunolocalization data presented here lack essential controls and there is no loss-of-function phenotype that has been identified.

We thank the reviewer for the critical review of our manuscript, and we greatly appreciate the comments that have helped us further strengthen our manuscript. Regarding the novelty of our work, we note that OsKCH2 is the first kinesin-14 motor to demonstrate processive minus-end-directed motility on single microtubules as a homodimer among all kinesin-14s that have been studied to date in plants, fungi and animals. We emphasize that the processive motility of OsKCH2 is quite distinct from the other two processive kinesin-14s Kar3 and KlpA, because the processive motility of Kar3 on single microtubules requires a *heterodimer* and the processive motility that KlpA achieves as a homodimer is *plus-end-directed*. This makes the current work a major conceptual advance on kinesin-14s in recent years. Its novelty and impact are clearly reflected in the comments from Reviewers #2 and #3 that are quoted here.

“The key finding here is the somewhat unexpected processivity of the rice OsKCH2 kinesin-14 minus-end directed motor protein. As the authors point out it is the first report of a processive plant kinesin-14, which could potentially compensate for the lack of cytoplasmic dynein from the plant lineage. This finding is not only relevant for plant cell biology, but provides insight for a general understanding of how the super-family of kinesin motors diversified their functions.” — Reviewer #2.

“...(This) is an extremely well prepared and well developed study with an important central discovery...” — Reviewer #3.

There are essential missing controls and other problems with this revised ms:

1. The authors have now decided that there are two upstream putative coiled coils, CC0

and CC1, not just one, although they continue to refer to CC1 as if it is the only upstream coiled coil. The coiled coil analysis they present is highly confusing, as it lacks positive and negative controls. The authors need to compare their program predictions with those for a known coiled coil, established using structural methods, and also with a protein whose known structure does not include a coiled coil to determine the validity of their coiled coil assignments based on the program predictions alone.

We thank the reviewer for suggesting us to run both positive and negative controls of the coiled-coil prediction by MARCOIL. In this revision, we have included the coiled-coil prediction of *human* kinesin-7 CENP-E(aa343-423) and yeast Pac11(aa25-87) (Supplementary Fig. 1). In either case, the coiled-coil prediction agrees extremely well with the experimental results (Jie et al., 2015; Phillips et al., 2016). Thus, the coiled-coil prediction of OsKCH2 is reliable.

The exact number of putative coiled-coils in OsKCH2 depends on the cutoff probability from the coiled-coil analysis. We used a relatively stringent cutoff probability of 90% in our original manuscript, which led to the assignment of two putative coiled-coils, and in the subsequent revision, we used a less stringent cutoff probability of 80%, which led to the assignment of three coiled-coils. We chose to name the first coiled-coil CC0 and thus were able to continue to use CC1 and CC2 for the two coiled-coils sandwiching the motor domain of OsKCH2. In doing so, we were hoping to: 1) minimize confusion that may arise for the reviewer due to these name changes, and 2) use the same set of names for the two coiled-coils that sandwich the motor domains of both OsKCH1 and OsKCH2. We have revised the entire manuscript to stress that CC1 is NOT the only upstream putative coiled-coil whenever necessary. For example, in page 6, we state that:

“We next characterized the motility of GFP-OsKCH2(289-767), a truncated motor-neck construct containing two putative coiled-coils CC1 and CC2 (Fig. 2a, b); it is worth emphasizing that GFP-OsKCH2(289-767) lacks the N-terminal CH domain, the other putative coiled-coil CC0 before CC1 and the C-terminus.”

It is not clear whether CC0 is a distinct coiled coil from CC1 - it appears to be the same coiled coil as CC1 with a hinge region. Moreover, CC2 in OsKCH2 has some of the features of a coiled coil, but the heptad repeat motif appears to deviate from a coiled coil in Sup Fig S5: a coiled coil is characterized by the hydrophobic residues in the two strands interacting with each other and it is not certain that this is the case for the region denoted as CC2. The region also does not appear to be a coiled coil in OsKCH1 by examination of the protein sequence. Note that the program should find the heptad repeat regions but the characteristics of the regions need to be analyzed further to determine whether the number of heptad repeats and the characteristics of the residues are those that are found in known coiled coils. From Fig S5, the authors may be incorrect in referring to this region as CC2. Their data indicating that the region does not dimerize the protein argues against the residues in this region adopting a coiled coil conformation, even if the predictive program they are using identifies it as such.

We recognize that the reviewer has made a number of valid points regarding the specifics of these putative coiled-coils in OsKCH2. For example, it is indeed unclear whether CC0 is a distinct coiled-coil from CC1; and for the putative “CC2” region in OsKCH2 (and also OsKCH1), it might be more appropriate not to refer to this region as something indicative of an authentic coiled-coil, as our data showed that the CC2 region in OsKCH2 lacks the ability to form an authentic coiled-coil, despite the predicted coiled-coil potential. However, we would like to point out that in the published works on KCH proteins, KCHs are commonly described as a class of novel kinesins with an internal kinesin motor flanked by two coiled-coiled regions (Dixit, 2012; Frey et al., 2009; Klotz and Nick, 2012; Reddy and Day, 2001; Richardson et al., 2006; Schneider and Persson, 2015). In our own writing, we felt compelled to follow this convention to refer to the downstream regions in both OsKCH2 and OsKCH1 as a putative coiled-coil (CC2).

They also should mention that CC0 is missing in the 289-767 construct - this should be made clear throughout the ms. What is the role of CC0?

We agree that it is important to clarify that the GFP-OsKCH2(289-767) construct does not contain the first putative coiled-coil, and have revised the manuscript to reflect that; see below and in the manuscript:

“We next characterized the motility of GFP-OsKCH2(289-767), a truncated motor-neck construct containing the two putative coiled-coils CC1 and CC2 (Fig. 2a, b); it is worth emphasizing that GFP-OsKCH2(289-767) lacks the N-terminal CH domain, the other putative coiled-coil CC0 before CC1 and the C-terminus.”

At present, the role of CC0 is not known, and we plan to investigate how CC0 and other nonmotor domains may regulate the motility of OsKCH2 in a separate study.

2. The phylogenetic analysis also lacks controls and essential information. The alignment used for the tree building is not shown and should be presented. Did the alignment include only the motor domain sequences and was it optimized? Did the tree search include an outgroup protein to determine whether all the proteins shown in the final tree grouped together? Were controls performed in which the protein sequences in the alignment were randomized and the tree search redone? Were other methods used to validate the groupings shown, e.g., pairwise comparisons of sequences to show that the proteins that group together are more closely related to one another than those in other groups? The protein names shown for the tree should be made shorter and the abbreviations and other information given in the figure legend to make the groupings in the tree easier to see at a glance.

We apologize for these omissions in our last revision. We have now included the sequence alignment of all the proteins for the tree building in this revision. Protein sequence alignment was carried out using the whole protein sequences because both the highly conserved motor domain and the characteristic residues outside the motor domain are important to this analysis. AtKinesin-12A was included in the analysis as an outgroup protein. Bootstrapping (n=100) was performed to estimate branch/clade supports.

Multiple programs (MUSCLE and T-Coffee for sequence alignment; maximum likelihood, parsimony and neighbor joining for creation of phylogenetic trees) gave similar results, in which KCH proteins were clustered together. A tree generated by the maximum likelihood method, representing the consensus tree, was presented here. Based on the reviewer's suggestion, we have used shortened protein names in the tree, whenever possible.

3. The antibodies against OsKCH2 are described as "monospecific" but there are no data showing that the antibodies recognize a specific epitope in OsKCH2. There are also no control experiments showing that the antibodies are specific for OsKCH2 and do not cross react with OsKCH1, another rice kinesin-14, or another protein. There are also no controls showing that the immunofluorescence is localized to the cell cortex surrounding the prophase nucleus - co-staining by a cortical antibody is needed to demonstrate this. The authors should also show the negative mutant cells, alongside the missing positive controls. If the authors want to conclude that the localization is to the preprophase band, they need to show a positive control that shows specific antibody staining to the preprophase band. The OsKCH2 staining in Fig 1b does not resemble the diagram of the PPB in Fig 1b. The biological significance of the OsKCH2 motor processivity is not at all clear, especially if it is redundant with one or more of the other rice KCH kinesins, as the authors suggest in their rebuttal as the reason for not obtaining a mutant phenotype.

First, we thank the reviewer for these comments. We have now removed "monospecific" in the description of the antibody, as more control experiments are needed for making such a definitive claim. We have also included the control immunofluorescence study of the *kch2* mutant cells. While KCH2 signal was clearly detected at the preprophase band in the wildtype cells, no KCH2 signal was detected at the preprophase band in the *kch2* mutant cells using the same antibody. Based on these observations, we conclude that the signal detected at the preprophase band by the antibody reflects the localization of OsKCH2 in prophase cells.

Second, we note that we also raised a second antibody against the C-terminal variable region of OsKCH2 (aa 752-1029) in rats. Using the second antibody, we detected immunofluorescence signals at the preprophase band that are consistent with the signals obtained with the rabbit antibody against aa 171-313 of OsKCH2. The localization data from the second antibody were not included in the manuscript to avoid repetitively presenting similar information.

Third, the diagram on the Left in our original Figure 1b was meant to facilitate readers who are not familiar with the preprophase band in land plants. When the preprophase band is observed from one side of the cell, it would leave the impression that the signal becomes more striking at the two edges of the cell. This is due to the nature of fluorescence imaging of aligned signals in vertical vs. horizontal orientations. The preprophase band is a ring of evenly distributed microtubule bundles encircling the prophase nucleus at the cell cortex. While our immunofluorescence data clearly show that OsKCH2 localizes to the preprophase band microtubules, our schematic diagram of the preprophase band likely is overly simplified. To avoid unintended confusion that may

arise due to our inability to draw a realistic diagram of the prophase band, we have removed the schematic diagram and referred the readers to those diagrams of the preprophase band that are properly drawn in other published works.

Fourth, we agree that the biological significance of the OsKCH2 motor processivity is still unknown. We do plan to carry out systematic studies in the future to uncover the potential physiological relevance of the novel OsKCH2 processive motility. However, due to functional redundancy that is frequently observed among plant genes, we argue that these experiments will be a substantial undertaking due to the lengthy growth period of this crop, and waiting for such data could seriously delay the publication of the breakthrough results reported here (that OsKCH2 is the first minus-end-directed processive kinesin-14 motor as a homodimer that interacts with actin filaments and exhibits a cell cycle-dependent localization pattern).

4. The bimodality of the velocity in Fig 1g is not apparent from the histogram, as also noted by another reviewer. What is the rationale for assigning two distinct velocities to the motor (other than the previous report for OsKCH1)? Why are these velocities for actin filament transport so much slower than the minus-end movement of the 289-767 OsKCH2 motor?

The velocity histogram of the actin transport data was fit to a bimodal distribution based on the following observations: (1) In a previous study (Walter et al., 2015), OsKCH1 was found to have two different velocities, and (2) In this study, two distinct actin transport velocities (fast and slow) were frequently observed for different actin filaments on the same microtubules. We agree that while the velocity histogram is skewed, the bimodality is not immediately clear. We have revisited the actin transport data thoroughly, and found that when multiple movies were acquired from a single slide, movies after the first two tended to over-sample the slow velocity mode, likely due to degradation of motor activities over time. We thus have repeated the actin transport experiments and used only the first two movies from each slide for velocity analysis. In doing so, we were able to generate a new velocity histogram with an apparent bimodal distribution; we have replaced Fig. 1e-g and Supplementary Movie 1 with the new dataset.

Presently, it is unclear why the actin transport is much slower than the minus-end-directed movement of individual OsKCH2. While this clearly warrants full investigation in the future, it is beyond the scope of the current study. We stress that the main conclusions of our current work do not hinge on the precise velocity distribution of actin transport by OsKCH2(1-767) and its magnitude. The main purpose of the *in vitro* actin transport experiments was to show that OsKCH2 not only simultaneously interacts with actin filaments and microtubules but also actively transports actin filaments along microtubules *in vitro*.

5. The authors should note that although their sedimentation data are consistent with OsKCH2 being a homodimer in solution, these data and their photobleaching assays do not rule out the possibility that two homodimers couple in the motility assays. Note that the photobleaching assays were performed under different conditions and geometry than

the motility assays. They state that they have demonstrated processive motility without clustering but they have not shown that the processive motility they observe is **not** due to clustering.

The reviewer seems to be concerned that the processive motility we observed for GFP-OsKCH2(289-767) could have been caused entirely by nonspecific clustering of the motor in the presence of microtubules, because the experiments we performed to assess dimerization were done in the absence of microtubules (sucrose gradient centrifugation and single-molecule photobleaching of motors stuck to glass surfaces). To address this concern, we carried out the following two new experiments:

First, we repeated all of the single-molecule photobleaching experiments with the kinesins bound to microtubules under the buffer condition identical to our single-molecule motility experiments, except that ATP was left out to allow the motors to tightly bind to the microtubule. The results of the new photobleaching dataset are consistent with the results in our previous revision, showing all the said “dimeric” constructs behaved like homodimers.

Second, we created a tetrameric GFP-OsKCH2(289-767)^T construct by inserting the coding sequence for a GCN4 parallel tetramer motif (Harbury et al., 1993) between GFP and OsKCH2 (Supplementary Fig 4.a). The same approach was recently used by the laboratories of Vale and Goshima to successfully make an artificial Kinesin14-VIb homotetramer (Jonsson et al., 2015). In contrast to the GFP-OsKCH2(289-767) dimer and similar to the Kinesin14-VIb homotetramer, GFP-OsKCH2(289-767)^T contained a high percentage of 3- and 4-step photobleaching processes (Supplementary Fig. 4c), indicating that OsKCH2(289-767)^T formed a homotetramer in solution. Most importantly, single-molecule experiments showed that OsKCH2(289-767)^T had a characteristic run-length of $16.5 \pm 3.0 \mu\text{m}$ (mean \pm s.e.m., $n = 287$, Supplementary Fig. 4e), >3.5 -fold longer than the dimer. The motility difference between GFP-OsKCH2(289-767) and the homotetrameric GFP-OsKCH2(289-767)^T is summarized below in Author Response Figure 1. Please note that for the run-length histogram of GFP-OsKCH2(289-767), we have changed its X-axis range to match that of GFP-OsKCH2(289-767)^T so that it is easier for the reviewer to see the dramatic difference in run length between these two constructs.

Taken together, the photobleaching results and the ultra-long run length demonstrated by the GFP-OsKCH2(289-767)^T tetramer argue strongly that the observed minus-end-directed processive motility for GFP-OsKCH2(289-767) is not due to inadvertent motor coupling and/or clustering on microtubules, but rather is an intrinsic behavior of the motor as a homodimer.

Author Response Figure 1. GFP-OsKCH2(289-767) exhibits drastically different motility from that of the artificial homotetramer GFP-OsKCH2(289-767)^I. **a**, Representative kymographs (Top), velocity histogram (Middle) and run-length histogram (Bottom) of individual GFP-OsKCH2(289-767) molecules on single microtubules. **b**, Representative kymographs (Top), velocity histogram (Middle) and run-length histogram (Bottom) of individual GFP-OsKCH2(289-767)^I molecules on single microtubules. Scale bars: 1 minute (vertical), and 5 μm (horizontal).

Reviewer #2 (Remarks to the Author):

In the revised manuscript, the authors further investigated the origin of processivity in

OsKCH2 by analysis of chimera between OsKCH2 and OsKCH1 and by mutation of a stretch of basic residues in CC2, expected to modulate the interaction between the kinesin and microtubules. Furthermore, the authors include *in vivo* co-localization analysis of OsKCH2 with the preprophase band and extend the manuscript by a phylogenetic tree and extend the protein domain analysis. In my opinion, these revisions adequately addressed the major concerns raised by the reviewers.

Reference:

Dixit, R. (2012). Putting a bifunctional motor to work: insights into the role of plant KCH kinesins. *New Phytol.* *193*, 543–545.

Frey, N., Klotz, J., and Nick, P. (2009). Dynamic bridges — a calponin-domain kinesin from rice links actin filaments and microtubules in both cycling and non-cycling cells. *Plant Cell Physiol.* *50*, 1493–1506.

Harbury, P.B., Zhang, T., Kim, P.S., and Alber, T. (1993). A switch between two-, three-, and four-stranded coiled coils in GCN4 leucine zipper mutants. *Science* *262*, 1401–1407.

Jie, J., Löhr, F., and Barbar, E. (2015). Interactions of yeast dynein with dynein light chain and dynactin: General implications for intrinsically disordered duplex scaffolds in multiprotein assemblies. *J. Biol. Chem.* *290*, 23863–23874.

Jonsson, E., Yamada, M., Vale, R.D., and Goshima, G. (2015). Clustering of a kinesin-14 motor enables processive retrograde microtubule-based transport in plants. *Nat. Plants* *1*, 15087.

Klotz, J., and Nick, P. (2012). A novel actin–microtubule cross-linking kinesin, NtKCH, functions in cell expansion and division. *New Phytol.* *193*, 576–589.

Phillips, R.K., Peter, L.G., Gilbert, S.P., and Rayment, I. (2016). Family-specific kinesin structures reveal neck-linker length based on Initiation of the coiled-coil. *J. Biol. Chem.* *291*, 20372–20386.

Reddy, A.S., and Day, I.S. (2001). Kinesins in the Arabidopsis genome: A comparative analysis among eukaryotes. *BMC Genomics* *2*, 2.

Richardson, D.N., Simmons, M.P., and Reddy, A.S. (2006). Comprehensive comparative analysis of kinesins in photosynthetic eukaryotes. *BMC Genomics* *7*, 18.

Schneider, R., and Persson, S. (2015). Connecting two arrays: the emerging role of actin-microtubule cross-linking motor proteins. *Front. Plant Sci.* *6*, 415.

Walter, W.J., Machens, I., Rafeian, F., and Diez, S. (2015). The non-processive rice kinesin-14 OsKCH1 transports actin filaments along microtubules with two distinct velocities. *Nat. Plants* *1*, 15111.